# If Optimizing for General Parameters in Chemistry is Useful, Why is it Hardly Done?

## Abstract

General parameters are highly desirable in the natural sciences — e.g., reaction conditions that enable high yields across a range of related transformations. This has a significant practical impact since those general parameters can be transfered to related tasks without the need for laborious and time-intensive re-optimization. While Bayesian optimization (BO) is widely applied to find optimal parameter sets for specific tasks, it has remained underused in experiment planning towards such general optima. In this work, we consider the real-world problem of condition optimization for chemical reactions to study whether performing generality-oriented BO can accelerate the identification of general optima, and whether these optima also translate to unseen examples. This is achieved through a careful formulation of the problem as an optimization over curried functions, as well as systematic benchmarking of generality-oriented strategies for optimization tasks on real-world experimental data. Empirically, we find that for generality-oriented optimization, simple optimization strategies that decouple parameter and task selection perform comparably to more complex ones, and that effective optimization is merely determined by an effective exploration of both parameter and task space.

## 1 Introduction

Identifying parameters that deliver satisfactory performance on a wide set of tasks, which we refer to as *general parameters*, is crucial for numerous real-world challenges. Examples are the identification of sensor settings that allow the sensor to measure accurately in different environments (Güntner et al., 2019), or the design of footwear that provides good performance for a range of people on different undergrounds (Promjun & Sahachaisaeree, 2012). A prominent example comes from the domain of chemical synthesis, where finding reaction conditions under which different starting materials can be reliably converted into the corresponding products, remains a critical challenge (Wagen et al., 2022; Prieto Kullmer et al., 2022; Rein et al., 2023; Betinol et al., 2023; Rana et al., 2024; Schmid et al., 2024). Such general conditions are of particular interest, *e.g.*, in the pharmaceutical industry, where thousands of reactions are carried out regularly, and optimizing each reaction is unfeasible (Wagen et al., 2022). While Bayesian Optimization (BO) is increasingly adopted within reaction optimization (Clayton et al., 2019; Shields et al., 2021; Guo et al., 2023; Tom et al., 2024), the vast majority of cases neglects generality considerations (Figure 1, left-hand side.)

This lack of consideration can be attributed to the fact that directly observing the generality of selected conditions is associated with largely increased experimental costs, as experimental evaluations on multiple substrates are required. Attempts at reducing the required number of experiments inevitably increase the complexity of the decision-making process. Thus, the usage of generality-oriented optimization in laboratories is hindered in the absence of appropriate decision-making algorithms. Here, generality-oriented optimization turns into a *partial monitoring scenario*, in which each condition can only be evaluated on a subset of all possible substrates. As a consequence, any iterative experiment planning algorithm needs to recommend both the condition and the substrate for the next experimental evaluation (Figure 1, right-hand side). Experimentally measuring the outcome of the recommended experiment corresponds to a partial observation of the generality objective, which needs to be taken into account when recommending the next experiment.

In the past two years, isolated studies have targeted the identification of general reaction conditions through variations of BO (Angello et al., 2022) and multi-armed bandit optimization (Wang

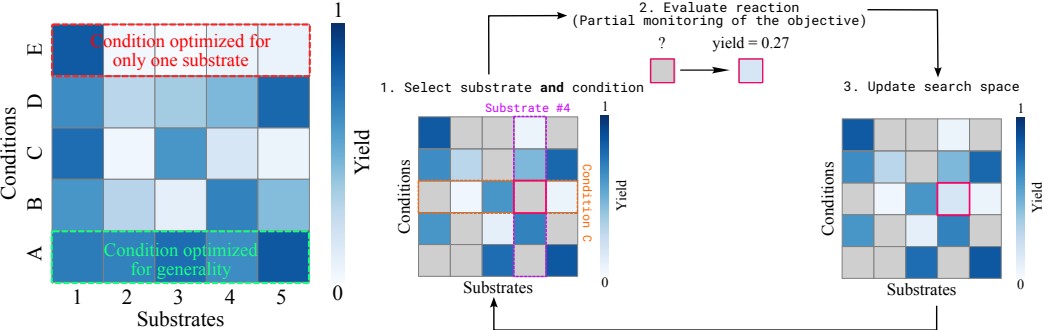

Figure 1: *Left:* While conditions can be optimized to maximize the reaction outcome for only one substrate (red), generality-optimized conditions provide a satisfactory reaction outcome for multiple substrates. *Right:* Optimization loop for generality-oriented optimization under partial monitoring.

et al., 2024). Concurrently, different algorithms have been proposed to optimize similarly structured problems, such as BO with expensive integrands (BOEI; Xie et al., 2012; Toscano-Palmerin & Frazier, 2018) and distributionally robust BO (DRBO; Bogunovic et al., 2018; Kirschner et al., 2020a). Despite these advances, generality-oriented optimizations are still not commonly performed in real-world experiments (see Section 2.2.4).This likely arises from the fact that the applicability and limitations of these algorithms are yet to be understood, which is crucial for their effective integration into real-world laboratory workflows (Tom et al., 2024).

For these reasons, we herein perform a systematic benchmark study into generality-oriented optimization. To obtain a problem flexibility required for real-world applications (Betinol et al., 2023), we formulate generality-oriented optimization as an optimization problem over curried functions. Further, we perform systematic benchmarks on various real-world chemical reaction optimization tasks.Specifically for the latter, we (i) confirm the expectation that optimization over multiple substrates leads to more general optima, and (ii) demonstrate that finding these optima effectively can be achieved through a highly explorative acquisition of the next conditions to evaluate.

In summary, our contributions are four-fold:

- Formulation of generality-oriented optimization as an optimization problem over a curried function.
- Expansion and adaptation of established reaction optimization benchmark tasks, improving their utility as benchmarks for generality-oriented BO.
- Evaluation of different optimization algorithms for identifying general optima.
- *CurryBO* as an open-source extension to *BoTorch* (Balandat et al., 2020) for generality-oriented optimization problems (noa).

## 2 FOUNDATIONS OF GENERALITY-ORIENTED BAYESIAN OPTIMIZATION

To formalize the generality-oriented optimization problem, we provide a principled outline by considering it as an extension of established global optimization approaches over curried functions. For clarity, we also discuss its distinction to different variations of global optimization, including multiobjective, multifidelity, and mixed-variable optimization.

### 2.1 GLOBAL OPTIMIZATION

Global black-box optimization is concerned with finding the optimum of an unknown objective function $f(\mathbf{x})$:

$$\hat{\mathbf{x}} = \underset{\mathbf{x} \in \mathcal{X}}{\operatorname{argmax}} f(\mathbf{x}) \tag{1}$$

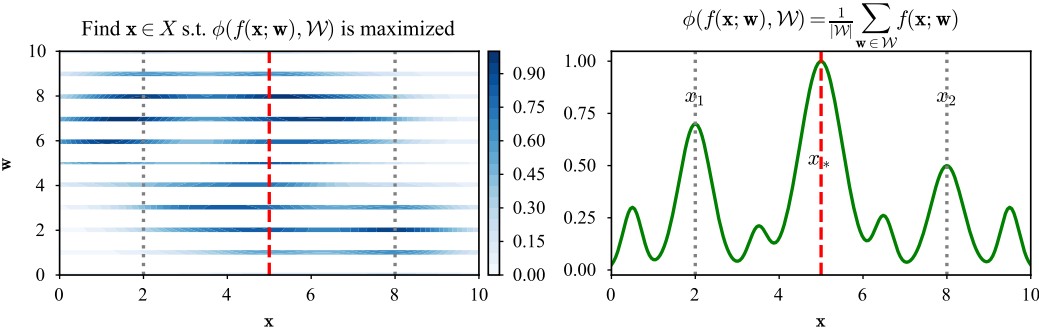

Figure 2: Conceptual overview of the generality-oriented optimization problem. *Left:* The function values across the joint space $\mathcal{X} \times \mathcal{W}$. *Right:* Mean aggregation applied to the function family $f(\mathbf{x}; \mathbf{w})$, that is obtained via currying of the joint space $\mathcal{X} \times \mathcal{W}$. The quantity $\phi\big(f(\mathbf{x}; \mathbf{w}), \mathcal{W}\big)$ constitutes the partially observable objective function, of which $\hat{\mathbf{x}} = \arg\max_{\mathbf{x} \in \mathcal{X}} \phi(\mathbf{x})$ is the optimum that should be identified.

Suppose $f(\mathbf{x})$ is a function that (a) is not analytically tractable, (b) is very expensive to evaluate, and (c) can only be evaluated without obtaining gradient information. In this scenario, BO has emerged as a ubiquitous approach for finding the global optimum $\hat{\mathbf{x}} \in \mathcal{X}$ in a sample-efficient manner (Garnett, 2023). The working principle of BO involves a probabilistic surrogate model $g(\mathbf{x})$ to approximate $f(\mathbf{x})$, which can be used to compute a predictive posterior distribution over $g$ under all previous observations $\mathcal{D} = \{(\mathbf{x}_i, f(\mathbf{x}_i)\}_{i=1}^k$. The most prominent choice for $p(g(\mathbf{x}) \mid \mathcal{D})$ are Gaussian processes (GPs; Rasmussen & Williams, 2006), with various types of Bayesian neural networks becoming increasingly popular in the past decade (Hernández-Lobato et al., 2017; Kristiadi et al., 2023; Li et al., 2024; Kristiadi et al., 2024). Based on the predictive posterior, an acquisition function $\alpha$ over the input space $\mathcal{X}$ is used to decide at which $\mathbf{x}_{\text{next}} \in \mathcal{X}$ the objective function should be evaluated next. Key to the success of BO is the implicit exploitation–exploration tradeoff in $\alpha$, which makes use of the posterior distribution $p(g(x) \mid \mathcal{D})$ (Močkus, 1975). Common choices of $\alpha$ are Upper Confidence Bound (UCB; Kaelbling, 1994a;b; Agrawal, 1995), Expected Improvement (EI; Jones et al., 1998), Knowledge Gradient (Gupta & Miescke, 1994; Frazier et al., 2008; 2009) or Thompson Sampling (TS; Thompson, 1933). The hereby selected $\mathbf{x}_{\text{next}}$ is evaluated experimentally, resulting in $f(\mathbf{x}_{\text{next}})$, and the described procedure is repeated until a satisfactory outcome is observed, or the experimentation budget is exhausted.

## 2.2 GLOBAL OPTIMIZATION FOR GENERALITY

### 2.2.1 PROBLEM FORMULATION

Extending the global optimization framework, we consider a black-box function $f : \mathcal{X} \times \mathcal{W} \to \mathbb{R}$ in joint space $\mathcal{X} \times \mathcal{W}$, where $\mathbf{x} \in \mathcal{X}$ can be continuous, discrete or mixed-variable and $\mathcal{W} = \{\mathbf{w}_i\}_{i=1}^n$ is a discrete parameter space of size $n$ (see Figure 2). Each evaluation of $f$ is expensive and does not provide gradient information. In the example of reaction condition optimization, $\mathbf{x}$ are conditions from the condition space $\mathcal{X}$, e.g. the temperature, and $\mathbf{w} \in \mathcal{W}$ the substrates (starting materials of a reaction) that are considered for generality-oriented optimization. Let curry be a currying operator on the second argument, i.e., $\text{curry}(f) : \mathcal{W} \to \big(\mathcal{X} \to \mathbb{R}\big)$. Then, for some $\mathbf{w} \in \mathcal{W}$, evaluating $\text{curry}(f)(\mathbf{w})$ yields a new function $f(\cdot\,; \mathbf{w}) : \mathcal{X} \to \mathbb{R}$, where $f(\mathbf{x}; \mathbf{w}) = f(\mathbf{x}, \mathbf{w})$. Importantly, these $f(\cdot\,; \mathbf{w}) : \mathcal{X} \to \mathbb{R}$ correspond to functions that can be evaluated experimentally (i.e. a reaction for a specific substrate as a function of conditions), even though evaluations are expensive. This allows us to describe all observable functions through an $n$-sized set $\mathcal{F} = \{f(\cdot\,; \mathbf{w}_i) : \mathcal{X} \to \mathbb{R}\}_{i=1}^n$. In the context of reaction condition optimization $\mathcal{F}$ consists of all functions that describe the reaction outcome for each substrate. Evaluation of a specific $f(\mathbf{x}_{\text{obs}}; \mathbf{w}_{\text{obs}})$ then corresponds to measuring the reaction outcome of a substrate (described by $\mathbf{w}_{\text{obs}}$) under specific reaction conditions $\mathbf{x}_{\text{obs}}$.

In generality-oriented optimization, the goal is to identify the optimum $\hat{\mathbf{x}} \in \mathcal{X}$ that is generally optimal across $\mathcal{W}$, meaning $\hat{\mathbf{x}}$ maximizes a user-defined generality metric over all $\mathbf{w} \in \mathcal{W}$ (see

---

**Algorithm 1** Generality-oriented Bayesian optimization

---

**Input:**

    Set of observable functions $\mathcal{F} = \{f(\cdot; \mathbf{w}_i) : \mathcal{X} \to \mathbb{R}\}_{i=1}^{n}$

    Initial dataset $\mathcal{D}_k = \left\{\mathbf{x}_j, \mathbf{w}_j, f(\mathbf{x}_j; \mathbf{w}_j)\right\}_{j=1}^{k}$

    Aggregation function $\phi\big(f(\mathbf{x}; \mathbf{w}), \mathcal{W}\big)$

    Surrogate model $g(\mathbf{x}, \mathbf{w})$ and acquisition policy $A$

    Budget $K$

1: **while** $k \leq K$ **do**
2:     Compute posterior distribution $p\big(g_k(\mathbf{x}, \mathbf{w}) \mid \mathcal{D}_k\big)$
3:     Acquire $\mathbf{x}_{k+1}, \mathbf{w}_{k+1} = A\Big(p\big(g_k(\mathbf{x}, \mathbf{w}) \mid \mathcal{D}_k\big), \phi\big(f(\mathbf{x}; \mathbf{w}), \mathcal{W}\big)\Big)$
4:     Observe $f(\mathbf{x}_{k+1}; \mathbf{w}_{k+1})$
5:     Update $\mathcal{D}_{k+1} = \mathcal{D}_k \bigcup \left\{(\mathbf{x_{k+1}}, \mathbf{w}_{k+1}, f(\mathbf{x}_{k+1}\mathbf{w}_{k+1}))\right\}$
6:     $k = k + 1$
7: **end while**

8: **return** $\hat{\mathbf{x}} = \underset{\mathbf{x} \in \mathcal{X}}{\mathrm{argmax}} \, \mathbb{E}\Big[p\big(\phi(\mathbf{x}) \mid \mathcal{D}_K\big) \mid \mathbf{x}\Big]$

---

Figure 2 for illustration). We refer to this generality metric as the *aggregation function $\phi$*:

$$\hat{\mathbf{x}} = \underset{\mathbf{x} \in \mathcal{X}}{\mathrm{argmax}} \, \phi(\mathbf{x}) := \underset{\mathbf{x} \in \mathcal{X}}{\mathrm{argmax}} \, \phi\big(f(\mathbf{x}; \mathbf{w}), \mathcal{W}\big) \qquad (2)$$

In the reaction optimization example, this corresponds to conditions (e.g. reaction temperature) that give e.g. the highest average yield over all considered substrates. In this scenario, the choice of $\phi$ is the mean $\phi(f(\mathbf{x}; \mathbf{w}), \mathcal{W}) = 1/|\mathcal{W}| \sum_{\mathbf{w} \in \mathcal{W}} f(\mathbf{x}; \mathbf{w})$. An alternative choice of $\phi$ could be the number of function values $\{f(\mathbf{x}; \mathbf{w}_i)\}_{i=1}^{n}$ above a user-defined threshold (Betinol et al., 2023). Further practically relevant aggregation functions are described in Appendix A.1.1.

While equation 2 appears like a standard global optimization problem over $\mathcal{X}$, evaluating $\phi(\mathbf{x})$ itself is intractable due to the aggregation over $\mathcal{W}$. Indeed, to evaluate $\phi(\mathbf{x})$ on a single $\mathbf{x}$, one must perform $n$-many expensive function evaluations to first obtain $\{f(\mathbf{x}; \mathbf{w}_i)\}_{i=1}^{n}$. Due to this intractability, ideally, the number of such function evaluations is minimized. Thus, this setting differs from the conventional global optimization problem, due to its *partial observation* nature: One can only compute $\phi(\mathbf{x})$ via a subset of observations $\{f(\mathbf{x}; \mathbf{w}_j)\}_{j=1}^{m}$ where $m < n$.

To maximize sample efficiency, an optimizer should always recommend a new pair $(\mathbf{x}_{\text{next}}, \mathbf{w}_{\text{next}})$ to evaluate next — in other words: $\phi(\mathbf{x}_{\text{next}})$ is only observed partially via a single evaluation of $f$, i.e., $m = 1$. Treating this in the conventional framework of BO, we can build a probabilistic surrogate model $g(\mathbf{x}_i; \mathbf{w}_i)$ from all $k$ available observations $\mathcal{D} = \{(\mathbf{x}_i, \mathbf{w}_i, f(\mathbf{x}_i; \mathbf{w}_i)\}_{i=1}^{k}$, referred to as $p\big(g_k(\mathbf{x}, \mathbf{w}) \mid \mathcal{D}\big)$. From the posterior distribution over $g$, a posterior distribution over $\phi$ can be estimated for any functional form of $\phi$ via Monte-Carlo integration (see Appendix A.1.2 for further details; Balandat et al., 2020).

Unlike the conventional BO case, we now need a specific acquisition policy $A$ to decide which $\mathbf{x} \in \mathcal{X}$ *and* $\mathbf{w} \in \mathcal{W}$ the aggregated objective function $\phi(\mathbf{x})$ should be partially evaluated. Note that $A$ plays an important role since it must respect the partial observability constraint. That is, it must also propose *a single* $\mathbf{w}$ at each BO step such that the general (over *all* $\mathbf{w}_i$'s) optimum $\hat{\mathbf{x}}$ is obtained in as few steps as possible. Given the pair $(\mathbf{x}_{k+1}, \mathbf{w}_{k+1})$, the aggregated objective $\phi(\mathbf{x}_{k+1})$ is partially observed, $\mathcal{D}$ is updated, and the discussed steps are repeated until the experimentation budget is exhausted. Eventually, owing to the partial monitoring scenario (Rustichini, 1999; Lattimore & Szepesvári, 2019; 2020), the final optimum after a budget of $k$ experiments, $\hat{\mathbf{x}}_k$, is returned as the $\mathbf{x} \in \mathcal{X}$ that maximizes the mean of the predictive posterior of $\phi$. A summary of this is provided in Algorithm 1.

### 2.2.2 ACQUISITION STRATEGIES TO SELECT $\mathbf{x}_{\text{NEXT}}$ AND $\mathbf{w}_{\text{NEXT}}$

As outlined above, the efficiency of generality-oriented optimization depends on the selection of $\mathbf{x}_{\text{next}}$ and $\mathbf{w}_{\text{next}}$. Given a posterior distribution $p\big(g_k(\mathbf{x}, \mathbf{w}) \mid \mathcal{D}\big)$, an aggregation function

$\phi\big(f(\mathbf{x}; \mathbf{w}), \mathcal{W}\big)$, any acquisition policy should determine $\mathbf{x}_{\text{next}}$ and $\mathbf{w}_{\text{next}}$, which formally requires optimization over $\mathcal{X} \times \mathcal{W}$. Assuming weak coupling between $\mathcal{X}$ and $\mathcal{W}$, we can formulate a sequential acquisition policy, as outlined in Algorithm 2. First, $\mathbf{x}_{\text{next}}$ is acquired by optimizing an $\mathbf{x}$-specific acquisition function $\alpha_x$ over the posterior over the aggregation function. Second, a $\mathbf{w}$-specific acquisition $\alpha_w$ is optimized over the posterior distribution at $\mathbf{x}_{\text{next}}$. Notably, in this setting, established one-step-lookahead acquisition functions can be used for both $\alpha_x$ and $\alpha_w$.

---

**Algorithm 2** Sequential Acquisition Strategy

**Input:**
      posterior distribution $p\big(g_k(\mathbf{x}, \mathbf{w}) \mid \mathcal{D}\big)$
      aggregation function $\phi\big(f(\mathbf{x}; \mathbf{w}), \mathcal{W}\big)$
      acquisition function $\alpha_x$
      acquisition function $\alpha_w$

1: compute posterior distribution $p\big(\phi(\mathbf{x}) \mid \mathcal{D}\big) = p\Big(\phi\big(g_k(\mathbf{x}, \mathbf{w}), \mathcal{W}\big) \mid \mathcal{D}\Big)$

2: acquire $\mathbf{x}_{\text{next}} = \underset{\mathbf{x} \in \mathcal{X}}{\operatorname{argmax}} \; \alpha_x\Big(p(\phi(\mathbf{x}) \mid \mathcal{D})\Big)$

3: acquire $\mathbf{w}_{\text{next}} = \underset{\mathbf{w} \in \mathcal{W}}{\operatorname{argmax}} \; \alpha_w\Big(p(g_k(\mathbf{x}_{\text{next}}, \mathbf{w}) \mid \mathcal{D})\Big)$

4: **return** $\mathbf{x}_{\text{next}}, \mathbf{w}_{\text{next}}$

---

However, the decoupling of $\mathcal{X}$ and $\mathcal{W}$ is a strong simplification, and identifying $\mathbf{x}_{\text{next}}$ and $\mathbf{w}_{\text{next}}$ formally requires a joint optimization over $\mathcal{X} \times \mathcal{W}$ (Algorithm 3). Such a joint optimization necessitates a two-step lookahead acquisition function $\alpha'$

$$\alpha'(\mathbf{x}_{k+1}, \mathbf{w}_{k+1}) = \alpha\left[\underset{\mathbf{x} \in \mathcal{X}}{\operatorname{argmax}} \; \alpha_{\text{final}}\left(p\Big(\phi(\mathbf{x}) \mid \mathcal{D}_{k+1}^*\Big)\right)\right] \tag{3}$$

where $\alpha$ is a classical one-step lookahead acquisition function, which is evaluated at $\mathbf{x}_{k+2} \in \mathcal{X}$ which maximizes the final acquisition function $\alpha_{\text{final}}$ (in our case: the posterior mean) over a fantasy posterior distribution $p\Big(\phi(\mathbf{x}) \mid \mathcal{D}_{k+1}^*\Big)$. This distribution is obtained by conditioning the existing posterior on a new fantasy observation at $(\mathbf{x}_{k+1}, \mathbf{w}_{k+1})$. An implementation of equation Equation (3) using Monte-Carlo integration is given in Algorithm 4.

$\mathbf{x}_{\text{next}}$ and $\mathbf{w}_{\text{next}}$ are then acquired by optimizing $\alpha'$ in the joint input space $\mathcal{X} \times \mathcal{W}$.

---

**Algorithm 3** Joint Acquisition Strategy

**Input:**
      posterior distribution $p\big(g_k(\mathbf{x}, \mathbf{w}) \mid \mathcal{D}\big)$
      aggregation function $\phi\big(f(x; w), \mathcal{W}\big)$
      two-step lookahead acquisition function $\alpha'$

1: compute posterior distribution $p\big(\phi(\mathbf{x}) \mid \mathcal{D}\big) = p\Big(\phi\big(g_k(\mathbf{x}, \mathbf{w}), \mathcal{W}\big) \mid \mathcal{D}\Big)$

2: acquire $\mathbf{x}_{\text{next}}, \mathbf{w}_{\text{next}} = \underset{x, w \in \mathcal{X} \times \mathcal{W}}{\operatorname{argmax}} \; \alpha'\Big(p(\phi(\mathbf{x}) \mid \mathcal{D})\Big)$

3: **return** $\mathbf{x}_{\text{next}}, \mathbf{w}_{\text{next}}$

---

### 2.2.3 DISTINCTION FROM EXISTING VARIANTS OF THE BO FORMALISM

Despite seeming similarities with *multiobjective*, *multifidelity*, and *mixed-variable* optimization, the generality-oriented approach describes a distinctly different scenario:

- In contrast to *multiobjective* optimization, here, we consider a single optimization objective, i.e. $\phi(\mathbf{x})$. However, this objective can only be partially observed. Whereas the overall

optimization problem aims to identify $\hat{\mathbf{x}} \in \mathcal{X}$, finding the next recommended observation requires a joint optimization over $\mathcal{X}$ and $\mathcal{W}$.

- In contrast to *multifidelity* BO, the functions parameterized by $\mathbf{w} \in \mathcal{W}$ do not correspond to the same objective with different fidelities. Rather, they are independent functions which all contribute equally to the objective function $\phi(\mathbf{x})$.

- Unlike *mixed-variable* BO (Daxberger et al., 2020), the goal of generality-oriented BO is not to find $(\mathbf{x}, \mathbf{w})$ that maximizes the objective in the *joint* space. Rather, the goal is to find the set optimum $\hat{\mathbf{x}}$ that maximizes $\phi\big(f(\mathbf{x}; \mathbf{w}), \mathcal{W}\big)$ over $f(\mathbf{x}; \mathbf{w})$. In the case of $\phi$ being a sum, this bears resemblance to maximizing the *marginal* over $\mathbf{x}$ (see Figure 2). Moreover, $\mathcal{X}$ can be continuous or discrete, thus, $\mathcal{X} \times \mathcal{W}$ can be a fully-discrete space.

### 2.2.4 RELATED WORKS

Similarly structured problems have been previously described, mostly for specific formulations of the aggregation function $\phi$. Most prominently, if $\phi$ contains a sum over all $f(\cdot; \mathbf{w}_i)$ with $\mathbf{w}_i \in \mathcal{W}$, this problem has been referred to as optimization of integrated response functions (Williams et al., 2000), optimizing an average over multiple tasks (Swersky et al., 2013), or optimization with expensive integrands (Toscano-Palmerin & Frazier, 2018). The latter work proposes a BO approach, including a joint acquisition over $\mathcal{X} \times \mathcal{W}$ with the goal of maximizing the value of information. In the framework discussed above, this corresponds to a joint optimization of a two-step lookahead expected improvement, and is included in our benchmark experiments as JOINT 2LA-EI. The scenario in which $\phi$ corresponds to the $min$ operation, i.e. the objective is $\min_{\mathbf{w} \in \mathcal{W}} f(\mathbf{x}; \mathbf{w})$, has been discussed as distributionally robust BO (Bogunovic et al., 2018; Kirschner et al., 2020a; Nguyen et al., 2020; Husain et al., 2023). While these works provide advanced algorithmic solutions for the respective optimization scenarios, our goal was to benchmark the applicability of such algorithms in real-life settings. Therefore, the formulation as optimization over curried functions provides a flexible framework that covers aggregation functions of arbitrary functional form, and the implementation of *CurryBO* allows for rapid integration with the *BoTorch* ecosystem.

In the chemical synthesis, the concept of "reaction generality" has been discussed on multiple occasions, given its enormous importance for accelerating molecular discovery (Wagen et al., 2022; Prieto Kullmer et al., 2022; Rein et al., 2023; Betinol et al., 2023; Rana et al., 2024; Gallarati et al., 2024; Schmid et al., 2024). The first example of actual generality-oriented optimization in chemistry has been reported by Angello et al. (2022), who describe a modification of BO, sequentially acquiring $\mathbf{x}_{\text{next}}$ via $\alpha_x = $ PI (Probability of Improvement) and $\mathbf{w}_{\text{next}}$ via $\alpha_w = $ PV (Posterior Variance). The authors demonstrate its applicability in automated experiments on Suzuki–Miyaura cross couplings. A similar algorithm as described in their work is evaluated herein as the SEQ 1LA-UCB-VAR strategy. Following an alternative strategy, Wang et al. (2024) recently formulated generality-oriented optimization as a multi-armed bandit problem, where each arm corresponds to a possible reaction condition. While their algorithm has been successful in campaigns with few possible reaction conditions, the necessity of sampling all conditions at the outset of a campaign renders its application impractical for a high number of discrete conditions or even continuous variables. The algorithm described in their work is evaluated herein as the BANDIT strategy.

Despite these recent advances, the applicability and limitations of these algorithmic approaches in real-life settings have remained unclear. Thus, our work provides a systematic benchmark over different generality-oriented optimization strategies, at the example of generality-oriented reaction optimization in chemistry.

Due to the partial monitoring nature of generality-oriented optimization, we want to highlight work that has been conducted on the partial monitoring case for bandits (Rustichini, 1999; Lattimore & Szepesvári, 2019; 2020). However, to the best of our knowledge, works in this field has mostly dealt with an information-theoretic approach towards optimally scaling algorithms. We refer the readers to select publications (Lattimore & Szepesvari, 2019; Kirschner et al., 2020b; Lattimore & Gyorgy, 2021; Lattimore, 2022). Comprehensive benchmark of different strategies in the early stages of an optimization has not been applied to generality-optimization for chemical benchmark tasks.

Table 1: Nomenclature and description of the benchmarked acquisition strategies and acquisition functions in the main text. Further acquisition functions are described in Table 2.

| Acquisition Strategy | Acquisition Function |
|---|---|
| SEQ 1LA: Sequential acquisition of $\mathbf{x}_{next}$ and $\mathbf{w}_{next}$, each using a one-step lookahead acquisition function. The final $\hat{\mathbf{x}}$ is selected greedily. | UCB: Upper confidence bound ($\beta = 0.5$). |
| SEQ 2LA: Sequential acquisition of $\mathbf{x}_{next}$ and $\mathbf{w}_{next}$, each using a two-step lookahead acquisition function. The final $\hat{\mathbf{x}}$ is selected greedily. | UCBE: Upper confidence bound ($\beta = 5$). |
| JOINT 2LA: Joint acquisition of $\mathbf{x}_{next}$ and $\mathbf{w}_{next}$ using a two-step lookahead acquisition function. The final $\hat{\mathbf{x}}$ is selected greedily. | EI: Expected Improvement. |
| BANDIT: Multi-armed bandit algorithm as implemented by Wang et al. (2024). | PV: Posterior Variance. |
| RANDOM: Random selection of the final $\hat{\mathbf{x}}$. | RA: Random acquisition. |

## 3 METHODS

### 3.1 EXPERIMENTAL BENCHMARK PROBLEMS

In our benchmarks, we consider four real-world chemical reaction problems stemming from high-throughput experimentation (HTE; Zahrt et al., 2019; Buitrago Santanilla et al., 2015; Nielsen et al., 2018; Stevens et al., 2022; Wang et al., 2024). Each problem evaluates the optimization of a chemically relevant reaction outcome (such as enantioselectivity $\Delta\Delta G^{\ddagger}$, yield, or starting material conversion), and contains an experimental dataset of substrates, conditions and measured outcomes.
Extensive analysis of the benchmark problems can be found in Appendix A.2. At this stage, it should be noted that, while widely used as such, the problems have not been designed as benchmarks for reaction condition optimization. To mitigate the well-known bias of HTE datasets towards high-outcome experiments (Strieth-Kalthoff et al., 2022; Beker et al., 2022), we additionally augment the search space to incorporate larger domains of low-outcome results using a chemically sensible expansion workflow (see Appendix A.2.2 for further details).

### 3.2 OPTIMIZATION ALGORITHMS

Using the benchmark problems outlined above, we perform systematic evaluations of multiple methods for the identification of general optima. In the main text, we discuss the acquisition strategies and functions for recommending the next data point $(\mathbf{x}_{next}, \mathbf{w}_{next})$ as shown in Table 1. We name each experiment according to the acquisition strategy used, followed by specifications of the used acquisition functions $\alpha_x$ and $\alpha_w$ or $\alpha$ for sequential and joint acquisitions, respectively. As an example, a sequential two-step lookahead acquisition strategy with an Upper Confidence Bound as $\alpha_x$ and Posterior Variance as $\alpha_w$, is referred to as SEQ 2LA-UCB-PV. Each strategy is evaluated under two different generality definitions: the *mean* and the *number-above-threshold* aggregation (threshold aggregation) functions described in Section 2.2.1 (see Appendix A.1.1 for further details).

In all BO experiments, we used a GP surrogate, as provided in *BoTorch* (Balandat et al., 2020), with the Tanimoto kernel from *Gauche* (Griffiths et al., 2023). Molecules were represented using Morgan Fingerprints (Morgan, 1965) with 1024 bits and a radius of 2, generated using RDKit (Landrum, 2023). For each experiment, we provide statistics over 30 independent runs, each performed over different substrates and initial conditions. Further baseline experiments are discussed in Appendix A.4. To ensure cross-task comparability, we calculate the GAP as a normalized, problem-independent optimization metric (GAP = $(y_i - y_0)/(y^* - y_0)$, where $y_i$ is the true generality of the recommendation at experiment $k$ and $y^*$ is the true global optimum; Jiang et al., 2020).

## 4 RESULTS AND DISCUSSION

To assess the utility of generality-oriented optimization, it is necessary to validate the transferability of these general optima to unseen spaces. Therefore, we commence our analysis by systematically investigating all benchmark tasks using exhaustive grid search. This analysis reveals that, with an

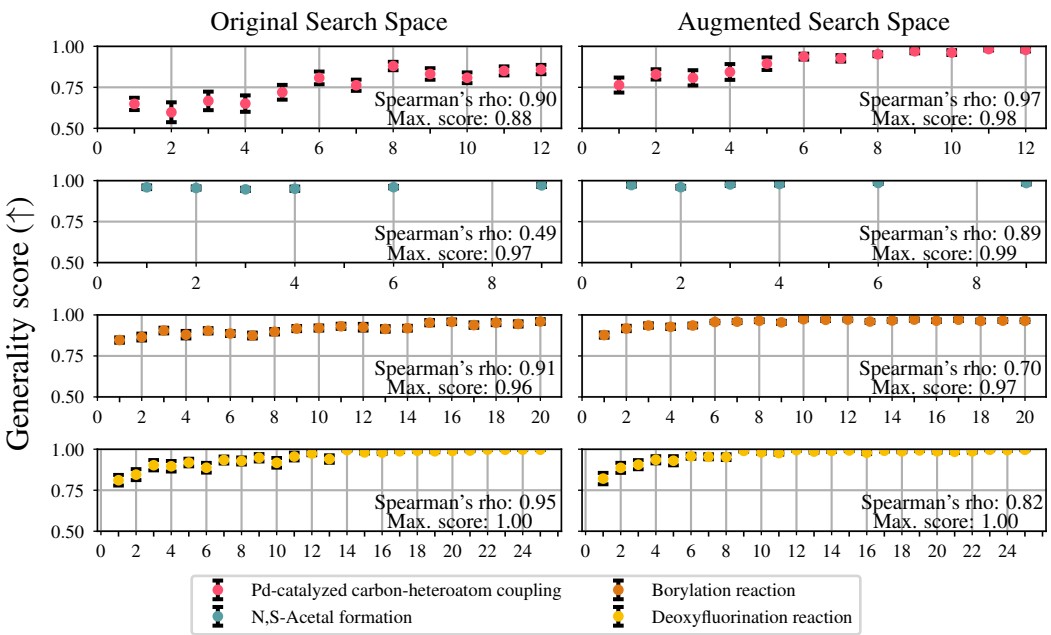

Figure 3: Normalized test-set generality score as determined by exhaustive grid search for the four benchmarks on the original (left) and augmented (right) problems for the mean aggregation. Average and standard error are taken from thirty different train/test substrates splits.

increasing number of substrates in $\mathcal{W}_{\text{train}}$ considered during optimization, the transferability of the found optima to a held-out test set $\mathcal{W}_{\text{test}}$ increases (Figure 3, left), as evidenced by Spearman's $\rho > 0$. While this finding is arguably unsurprising, and merely confirms a common assumption in the field (Wagen et al., 2022), it indicates possible caveats concerning the use of the non-augmented problems as benchmarks for generality-oriented optimization: Even with larger sizes of $\mathcal{W}_{\text{train}}$, the found optima did not consistently lead to optimal outcomes on the corresponding test sets. In contrast, we find that on the augmented benchmark tasks, which are more reflective of experimental reality, transferability of the identified optima to a held-out $\mathcal{W}_{\text{test}}$ is significantly improved. Notably, these observations are not limited to the definition of generality as the average over all $\mathbf{w} \in \mathcal{W}$, but remain valid for further aggregation functions on a majority of benchmarks (see Appendix A.6.1). These findings underline that – especially in "needle in a haystack scenarios" – generality-oriented optimization is indeed necessary for finding transferable optima. Most importantly, such scenarios apply to real-world reaction optimization, where for most reactions, the majority of possible conditions do not lead to observable product quantities. This re-emphasizes the need for benchmark problems that reflect experimental reality.

Having established the utility of generality-oriented optimization, we set out to perform a systematic benchmark of how to identify those optima using iterative optimization under partial objective monitoring. In the first step, we evaluate those approaches that have been developed in the context of reaction optimization (Angello et al., 2022; Wang et al., 2024) on two practically relevant aggregation functions, the mean and threshold aggregation (Appendix A.1.1). As a summary, Figure 4 shows the optimization trajectories of these different algorithms averaged across all augmented benchmark problems.. Overall, we find that the BO-based SEQ 1LA-UCB-PV acquisition strategy, as outlined by Angello et al. (2022), shows faster optimization performance compared to other algorithms used in the chemical domain. In particular, it significantly outperforms the BANDIT algorithm proposed by Wang et al. (2024), which can be attributed to the necessity of evaluating each $\mathbf{w} \in \mathcal{W}_{\text{train}}$ at the outset of each campaign, tying up a notable share of the experimental budget Assuredly, both proposed methods readily outperform the two random baselines RANDOM and SEQ 1LA-RA-RA.

Inspired by these observations, we perform a deeper investigation into the BO approaches formalized in Section 2.2. Initially, different options of the sequential strategy of acquiring $\mathbf{x}_{\text{next}}$ and $\mathbf{w}_{\text{next}}$ are

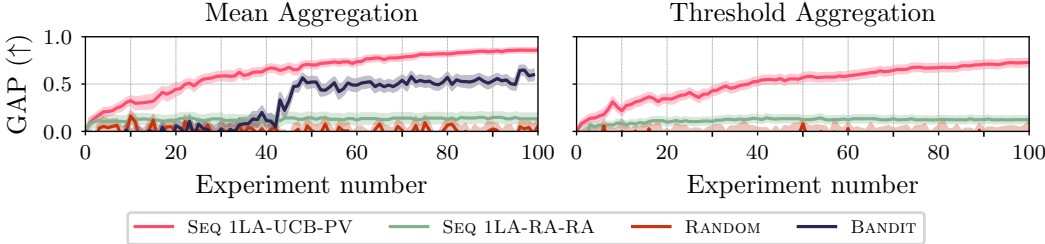

Figure 4: Optimization trajectories of different algorithms for generality-oriented optimization previously reported in the chemical domain. The trajectories are averaged over all augmented benchmark problems. Note that the BANDIT algorithm is incompatible with the threshold aggregation function.

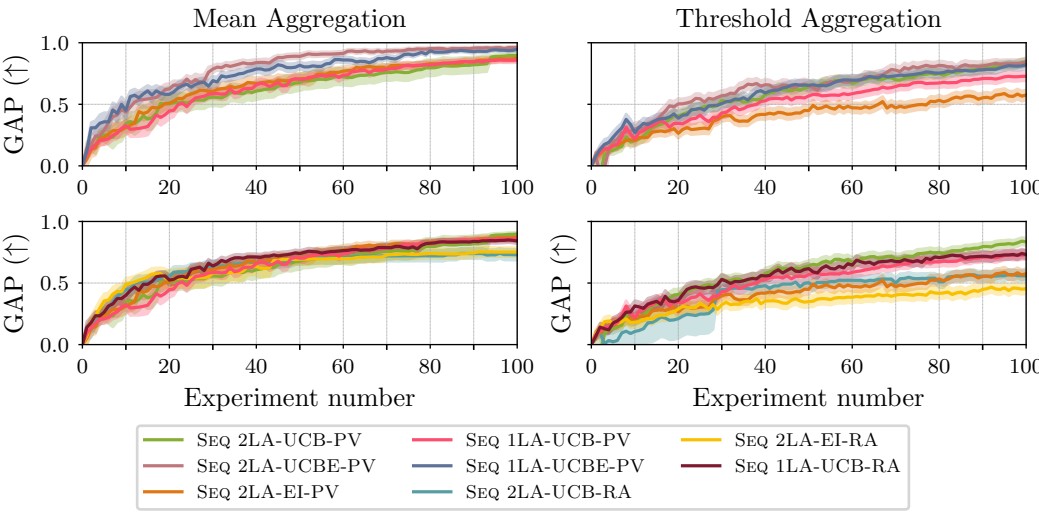

Figure 5: Optimization trajectories using sequential acquisition strategies. The top row shows the variation of $\alpha_x$, while the bottom row shows the variation of $\alpha_w$). Trajectories are averaged over four augmented benchmark problems.

evaluated. For this purpose, we compare multiple acquisition functions $\alpha_x$ for selecting $\mathbf{x}_{\text{next}} \in \mathcal{X}$, as formalized in Appendix A.4 and Section 3.2. Overall, the empirical results (Figure 5, top half) indicate largely similar optimization behavior for the different $\alpha_x$. However, it can be observed that a higher degree of exploration has a positive effect on optimization performance, e.g., when comparing the baseline method SEQ 1LA-UCB-PV ($\alpha_x$: UCB with $\beta = 0.5$) with the more exploratory SEQ 1LA-UCBE-PV ($\alpha_x$: UCB with $\beta = 5.0$). While systematic investigations into the generalizability of this finding are ongoing, we hypothesize that it can be attributed to the partial monitoring scenario, which leads to larger predictive uncertainties, and therefore less efficient exploitation. Surprisingly, the use of two-step-lookahead acquisition functions for $\alpha_x$, which should conceptually be well-suited for the partial monitoring scenario (Section 2.2.2), did not lead to significant improvements compared to their one-step-lookahead counterparts (e.g., comparing SEQ 1LA-UCB-PV with SEQ 2LA-UCB-PV and SEQ 2LA-EI-PV). Yet, the trend that more exploratory $\alpha_x$ lead to improved optimization behavior can also be observed for two-step-lookahead acquisition functions. However, we find that, especially for the threshold aggregation function (Figure 5), Expected Improvement (EI) shows significantly decreased optimization performance, which may be attributed to the partial monitoring scenario, and the resulting uncertainty in estimating the current optimum.

Similarly, we observe only a small influence of the choice of $\alpha_w$ (Figure 5, bottom half). In particular, an uncertainty-driven acquisition of $\alpha_w$, as used by Angello et al. (2022), shows only slightly

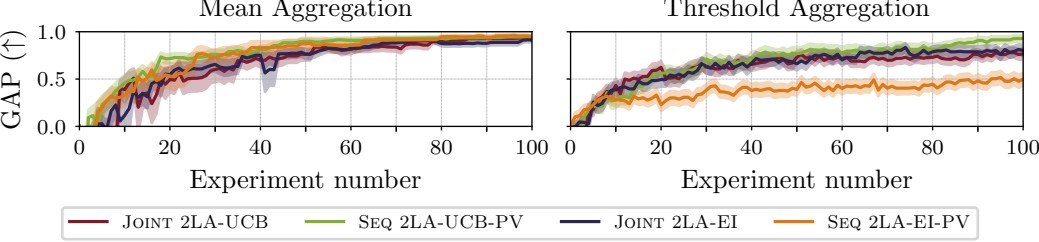

Figure 6: Optimization trajectories using sequential and joint two-step lookahead acquisition strategies. Note that, owing to computational cost constraints, the trajectories are only averaged over the N,S-Acetal formation and Deoxyfluorination reaction augmented benchmark problems.

improved optimization performance over a fully random acquisition of $\mathbf{w}_{\text{next}}$ (compare SEQ 1LA-UCB-PV and SEQ 1LA-UCB-RA). Notably, the difference becomes more pronounced for two-step lookahead acquisition policies (SEQ 2LA-UCB-PV and SEQ 2LA-UCB-RA). These findings indicate that, in the partial monitoring scenario, predictive uncertainties are not used effectively in *myopic* decision making, but their accurate propagation can improve *hyperopic* decisions. However, in the case of sequentially acquiring $\mathbf{x}_{\text{next}}$ and $\mathbf{w}_{\text{next}}$, this ability to effectively harness uncertainties for $\alpha_w$ does not lead to empirical performance improvements over the one-step lookahead acquisition policies. This could be attributed to the decoupling of $\mathcal{X}$ and $\mathcal{W}_{\text{train}}$.

Therefore, we eventually benchmark acquisitions strategies that recommend $\mathbf{x}_{\text{next}}$ and $\mathbf{w}_{\text{next}}$ through a joint optimization over $\mathcal{X} \times \mathcal{W}_{\text{train}}$, as originally proposed by Toscano-Palmerin & Frazier (2018) in the context of BO with expensive integrands. Figure 6 shows a comparison of different joint acquisition strategies to the sequential strategy discussed above. Empirically, we find that jointly optimizing for $\mathbf{x}_{\text{next}}$ and $\mathbf{w}_{\text{next}}$ does not lead to improved optimization performance, both when using EI and UCB as the acquisition function. However, we find that, in the case of joint acquisition, the discrepancies between EI and UCB that are observed in the sequential case, are no longer present, showcasing that the algorithm proposed by Toscano-Palmerin & Frazier (2018) can be applied in this context. However, given the increased computational cost of joint optimization, our empirical findings suggest that the algorithmically simpler sequential acquisition strategy with one-step lookahead acquisition functions is well-suited for generality-oriented optimization for chemical reactions, and performs on par with more advanced algorithmic approaches.

## 5 CONCLUSION

In this work, we extend global optimization frameworks to the identification of general and transferable optima, exemplified by the real-world problem of chemical reaction condition optimization. Systematic analysis of common reaction optimization benchmarks supports the hypothesis that optimization over multiple related tasks can yield more general optima, particularly in scenarios with a low the density of high-outcome experiments across the search space. We provide augmented versions of these benchmarks to reflect these real-life considerations. For BO aimed at identifying general optima, we find that a simple and cost-effective strategy —— sequentially optimizing one-step-lookahead acquisition functions over $\mathcal{X}$ and $\mathcal{W}$ — is well-suited, and performs on par with more complex policies involving two-step lookahead acquisition. Our analyses indicate that the choice of explorative acquisition function for sampling $\mathcal{X}$ is the most influential factor in achieving successful generality-oriented optimization, likely due to the partial optimization nature of the problem. While our findings mark an important step towards applying generality-oriented optimization in chemical laboratories, they also highlight the continued need for benchmark problems that accurately reflect real-world scenarios (Liang et al., 2021). We believe that such benchmarks, along with systematic evaluations of chemical reaction representations, are essential for a principled usage of generality-oriented optimization. Building on our results, we anticipate that generality-oriented optimization will see increasing adoption in chemistry and beyond, contributing to the development of more robust, applicable and sustainable reactions.

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

# A APPENDIX

## A.1 BAYESIAN OPTIMIZATION FOR GENERALITY

### A.1.1 AGGREGATION FUNCTIONS

The aggregation function is a user-defined property that determines how the "set optimum" is calculated across objective functions. Through the choice of the set optimum, prior knowledge and preferences about the specific optimization problem at hand can be included. In this work, the following aggregation functions are evaluated:

**Mean Aggregation**

$$\phi(f(\mathbf{x}; \mathbf{w}), \mathcal{W}) = \frac{1}{|\mathcal{W}|} \sum_{\mathbf{w} \in \mathcal{W}} f(\mathbf{x}; \mathbf{w}) = \frac{1}{n} \sum_{i=1}^{n} f(\mathbf{x}; \mathbf{w}_i) \tag{4}$$

**Threshold Aggregation**

$$\phi(f(\mathbf{x}; \mathbf{w}), \mathcal{W}) = \sum_{\mathbf{w} \in \mathcal{W}} \sigma(f(\mathbf{x}; \mathbf{w}) - f_{\text{thr}}) = \sum_{i=1}^{n} \sigma(f(\mathbf{x}; \mathbf{w}_i) - f_{\text{thr}}) \tag{5}$$

Conceivably, other aggregation functions also have practical use-cases, for example:

**Mean Squared Error (MSE) Aggregation**

$$\phi(f(\mathbf{x}; \mathbf{w}), \mathcal{W}) = -\frac{1}{|\mathcal{W}|} \sum_{\mathbf{w} \in \mathcal{W}} (f_{\text{opt}}(\mathbf{x}; \mathbf{w}) - f(\mathbf{x}; \mathbf{w}))^2 = -\frac{1}{n} \sum_{i=1}^{n} (f_{\text{opt},i} - f(\mathbf{x}; \mathbf{w}_i))^2 \tag{6}$$

**Minimum Aggregation**

$$\phi(f(\mathbf{x}; \mathbf{w}), \mathcal{W}) = \min_{\mathbf{w}_i \in \mathcal{W}} f(\mathbf{x}; \mathbf{w}_i) \tag{7}$$

The above definitions assume that all $f(\mathbf{x}; \mathbf{w}_i)$ have the same range, and that the optimization problem is formulated as maximization problem.

### A.1.2 ACQUISITION FUNCTIONS AND THE SAMPLE AVERAGE APPROXIMATION

For the evaluation of posterior distributions, and the calculation of acquisition function values, we use the sample-average approximation, as introduced by Balandat et al. (2020). From a posterior distribution at time point $k$, $p(g_k(\mathbf{x}))$, $M$ posterior samples $\zeta_m(\mathbf{x}) \sim p(g_k(\mathbf{x}))$ are drawn. These posterior samples can be used to estimate the posterior distribution, and to calculate acquisition function values as expectation values $\mathbb{E}_M$ over all $M$ samples.

Herein, we use the following common acquisition functions:

- Upper Confidence Bound: $\text{UCB}(\mathbf{x}) = \mathbb{E}_M(\zeta_m(\mathbf{x})) + \beta \cdot \mathbb{E}_M(\zeta_m(\mathbf{x}) - \mathbb{E}_M(\zeta_m(\mathbf{x})))$.

- Expected Improvement: $\text{EI}(\mathbf{x}) = \mathbb{E}_M(\zeta_m(\mathbf{x}) - f^*)$, where $f^*$ is the best value observed so far.

- Posterior Variance: $\text{PV}(\mathbf{x}) = \mathbb{E}_M(\zeta_m(\mathbf{x}) - \mathbb{E}_M(\zeta_m(\mathbf{x})))$.

- Random Selection, where the acquisition function value is a random number.

Moreover, we evaluate the optimization performance using a primitive implementation of two-step lookahead acquisition functions $\alpha^*$ (see Algorithm 4). The acquisition function value of $\alpha^*$ at a location $\mathbf{x_0}$ is estimated as follows: For each of the $M$ posterior samples $\zeta_m(\mathbf{x_0}) \sim p(g_k(\mathbf{x_0}))$, a fantasy posterior distribution $p'(\phi(g_{k+1}(\mathbf{x_0})))$ is generated by conditioning the posterior on the

new observation $(\mathbf{x_0}, \zeta_M(\mathbf{x_0}))$ and aggregation. From this fantasy posterior distribution, the values of the inner acquisition function $\alpha_m$ can be computed and optimized over $\mathbf{x} \in \mathcal{X}$. The final value of the two-step lookahead acquisition function is returned as $\alpha^*(\mathbf{x}_0) = \frac{1}{M} \sum_{m=1}^{M} \alpha_m$.

---

**Algorithm 4** Two-step lookahead acquisition function using the sample average approximation.

**Input:**
    input space $\mathcal{X}$
    location $\mathbf{x_0}$ at which to evaluate the two-step lookahead acquisition function
    aggregation function $\phi(f(\mathbf{x}; \mathbf{w}), \mathcal{W})$
    posterior distribution $p(g_k(\mathbf{x}) \mid \mathcal{D})$
    one-step lookahead acquisition function $\alpha(\mathbf{x})$

1: draw $M$ posterior samples $\zeta_m(\mathbf{x_0}) \sim p(g_k(\mathbf{x_0}))$
2: empty set of fantasy acquisition function values $\mathcal{A} = \{\}$
3: **for** $m = 1, \ldots, M$ **do**
4:     compute fantasy posterior $p'(\mathbf{x}) = p\Big(\phi\big(g_{k+1}(\mathbf{x}) \mid (\mathcal{D} \cup (\mathbf{x_0}, \zeta_m(\mathbf{x_0})))\big)\Big)$
5:     optimize one-step-lookahead acquisition function $\alpha_m = \max_{\mathbf{x} \in \mathcal{X}} \alpha(p'(\mathbf{x}))$
6:     update $\mathcal{A} = \mathcal{A} \cup \{\alpha_m\}$
7: **end for**
8: **return** $\alpha^*(\mathbf{x}_0) = \frac{1}{M} \sum_{m=1}^{M} \alpha_m$

---

### A.1.3 BENCHMARKED OPTIMIZATION STRATEGIES FOR SELECTING $\mathbf{x}_{\text{NEXT}}$ AND $\mathbf{w}_{\text{NEXT}}$

Herein, we outline the use of the benchmarked optimization strategies for generality-oriented optimization. The discussed optimization strategies describe different variations of how to pick the next experiments $\mathbf{x}_{\text{next}}$ and $\mathbf{w}_{\text{next}}$.

Following the SAA (Balandat et al., 2020) outlined above, we estimate the predictive posterior distribution $p(\phi(\mathbf{x}) \mid \mathcal{D})$ as follows: For each $\mathbf{w}_i \in \mathcal{W}$, $M$ (typically $M = 512$ for one-step lookahead strategies and $M = 3$ for two-step lookahead strategies to reduce computational costs) samples $\zeta_{im}(\mathbf{x}) \sim p(g_k(\mathbf{x}, \mathbf{w}_i))$ are drawn from the posterior distribution of the surrogate model. Aggregating over all $\mathbf{w}_i$ yields $M$ samples $\zeta_m(\mathbf{x}) \sim p(\phi(\mathbf{x}) \mid \mathcal{D})$ from the posterior distribution over $\phi(\mathbf{x})$, which can be used for calculating the acquisition function values using the sample-based acquisition function logic, as described in Appendix A.1.2. With this, we implement and benchmark the acquisition policies in Table 2.

The sequential acquisition is described in Algorithm 2 and refers to a strategy in which $\mathbf{x}_{\text{next}}$ and $\mathbf{w}_{\text{next}}$ are selected sequentially. In the first step, $\mathbf{x}_{\text{next}}$ is selected by optimizing an $\mathbf{x}$-specific acquisition function $\alpha_x$ over $\mathbf{x} \in \mathcal{X}$. With the selected $\mathbf{x}_{\text{next}}$ in hand, $\mathbf{w}_{\text{next}}$ is then selected by optimizing an independent, $\mathbf{w}$-specific acquisition function over $\mathbf{w} \in \mathcal{W}$. With $\alpha_x = \text{PI}$ (Probability of Improvement) and $\alpha_w = \text{PV}$, this would correspond to the strategy described in (Angello et al., 2022). In contrast, the joint acquisition, as outlined in Algorithm 3, refers to a strategy in which $\mathbf{x}_{\text{next}}$ and $\mathbf{w}_{\text{next}}$ are selected jointly through optimization of a two-step lookahead acquisition function (see Algorithm 4 and Appendix A.1.2).

### A.2 BENCHMARK PROBLEM DETAILS

#### A.2.1 ORIGINAL BENCHMARK PROBLEMS

Four chemical reaction benchmarks have been considered in this work: Reactant conversion optimization for Pd-catalyzed C–heteroatom couplings (Buitrago Santanilla et al., 2015), enantioselectivity optimization for a N,S-Acetal formation (Zahrt et al., 2019), yield optimization for a borylation reaction (Stevens et al., 2022; Wang et al., 2024) and yield optimization for deoxyfluorination reaction (Nielsen et al., 2018; Wang et al., 2024). Since it has been well-demonstrated that these problems can be effectively modeled by regression approaches (Zahrt et al., 2019; Ahneman et al., 2018; Sandfort et al., 2020), we trained a random forest regressor on each dataset, which was used as

Table 2: Nomenclature and description of the all benchmarked acquisition strategies and acquisition functions, as discussed in the main text and the Appendix. Each experiment is named according to the acquisition strategy used, followed by specifications of the used acquisition functions $\alpha_x$ and $\alpha_w$ or $\alpha$ for sequential and joint acquisitions, respectively. As an example, a sequential two-step lookahead acquisition strategy with an Upper Confidence Bound as $\alpha_x$ and Posterior Variance as $\alpha_w$, is referred to as SEQ 2LA-UCB-PV.

| Acquisition Strategy | Acquisition Function |
|---|---|
| SEQ 1LA: Sequential acquisition of $\mathbf{x}_{next}$ and $\mathbf{w}_{next}$, each using a one-step lookahead acquisition function. The final $\hat{\mathbf{x}}$ is selected greedily. | UCB: Upper confidence bound ($\beta = 0.5$). |
| SEQ 2LA: Sequential acquisition of $\mathbf{x}_{next}$ and $\mathbf{w}_{next}$, each using a two-step lookahead acquisition function. The final $\hat{\mathbf{x}}$ is selected greedily. | UCBE: Upper confidence bound ($\beta = 5$). |
| JOINT 2LA: Joint acquisition of $\mathbf{x}_{next}$ and $\mathbf{w}_{next}$ using a two-step lookahead acquisition function. The final $\hat{\mathbf{x}}$ is selected greedily. | EI: Expected Improvement. |
| BANDIT: Multi-armed bandit algorithm as implemented by Wang et al. (2024). | PV: Posterior Variance. |
| RANDOM: Random selection of the final $\hat{\mathbf{x}}$. | RA: Random acquisition. |
| | SINGLE: Selection of the same substrate ($\mathbf{w}$) for every iteration. |
| | COMPLETE: Selection of every substrate (i.e. every $\mathbf{w} \in \mathcal{W}$) for a selected $\mathbf{x}_{next}$. |

the ground truth for all benchmark experiments (Häse et al., 2021). In the following, the benchmark problems are described briefly.

**Pd-catalyzed carbon-heteroatom coupling**

The Pd-catalyzed carbon-heteroatom coupling benchmark is concerned with the reaction of different nucleophiles with 3-bromopyridine (Figure 7). In total, 16 different nucleophiles were tested in a nanoscale high-throughput experimentation platform. As reaction conditions, bases (six different bases) and catalysts (16 different catalysts) were varied. In total, the benchmark consists of 1536 different experiments, for which the conversion is reported.

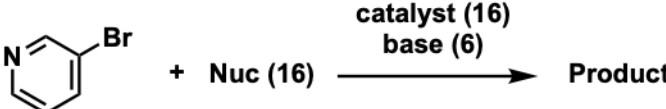

Figure 7: Reaction diagram of the Pd-catalyzed carbon-heteroatom coupling, where 3-bromopyridine reacts with a nucleophile. Reaction conditions include a catalyst and a base. The numbers indicate the amount of different species in the benchmark.

The average conversion is $2.05\%$, whereas the maximum conversion is $39.81\%$ (Figure 8). The average of the average conversion of each condition is $2.05\%$, while the maximum of the average conversion of the conditions is $7.60\%$ (Figure 8). The catalyst-base combination with the highest average conversion is shown in Figure 8.

With respect to the threshold aggregation function, the chosen threshold was $7.50\%$. The average number of substrates with a conversion above this threshold are $1.615$, while the maximum number of substrates is 7 (Figure 9). The catalyst-base combination with the highest number of substrates with a conversion above the threshold is the same as shown in Figure 8.

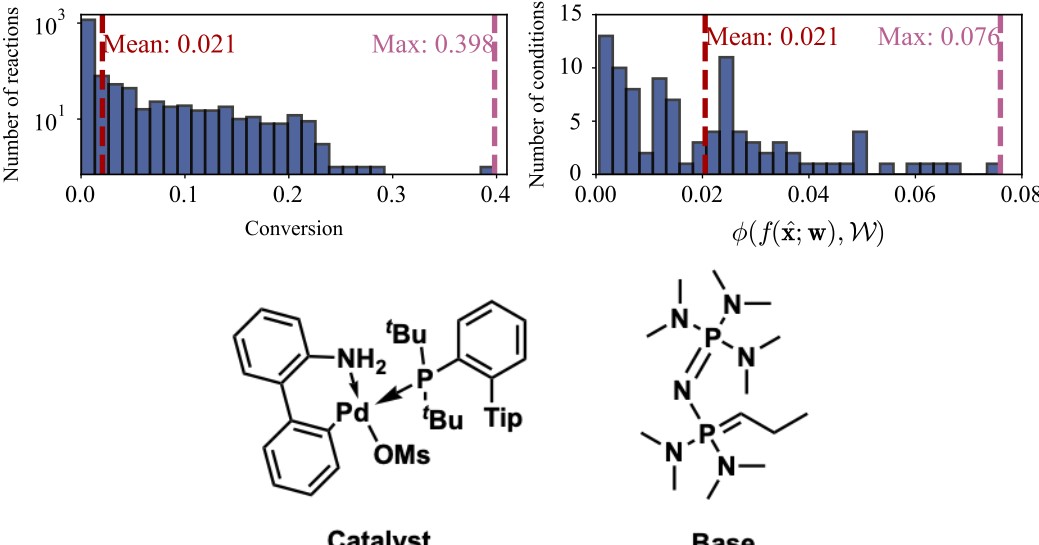

Figure 8: Top left: Distribution of the conversion for the Pd-catalyzed carbon-heteroatom coupling in the original benchmark. Top right: Distribution of the average conversion for each catalyst-base combination for the Pd-catalyzed carbon-heteroatom coupling in the original benchmark. Bottom: Catalyst-base combination with the highest average conversion in the original benchmark. Tip = 2,4,6-triisopropylphenyl.

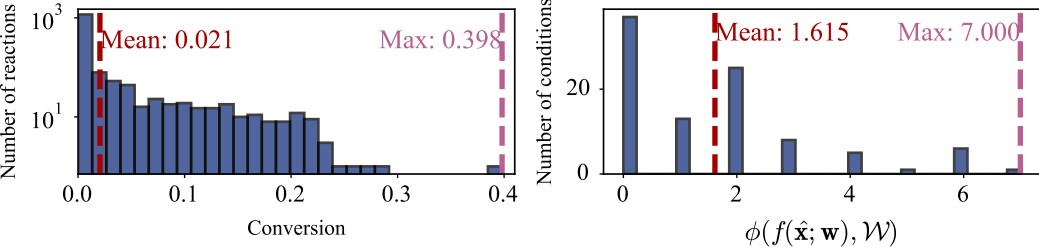

Figure 9: Left: Distribution of the conversion for the Pd-catalyzed carbon-heteroatom coupling in the original benchmark. Right: Distribution of the number of substrates with a conversion above the specified threshold for each catalyst-base combination for the Pd-catalyzed carbon-heteroatom coupling in the original benchmark.

**N,S-Acetal formation**

The N,S-Acetal formation benchmark is concerned with the nucleophilic addition of different thiols to imines, catalyzed by chiral phosphoric acids (CPAs) (see Figure 10). In total, five different imines and five different thiols were tested in manual experiments. As reaction conditions, 43 different CPA catalysts were considered. In total, the benchmark consists of 1075 different experiments, for which $\Delta\Delta G^{\ddagger}$, as a measure of the enantioselectivity, is reported.

The average $\Delta\Delta G^{\ddagger}$ is 0.988 kcal/mol, whereas the maximum $\Delta\Delta G^{\ddagger}$ is 3.135 kcal/mol (see Figure 11). The average of the average $\Delta\Delta G^{\ddagger}$ for each condition is 0.988 kcal/mol, while the maximum of the average $\Delta\Delta G^{\ddagger}$ for all conditions is 2.395 kcal/mol (see Figure 11). The catalyst with the highest average $\Delta\Delta G^{\ddagger}$ is shown in Figure 11.

With respect to the threshold aggregation function, the chosen threshold was 2.0 kcal/mol. The average number of substrates with $\Delta\Delta G^{\ddagger}$ above this threshold are 1.907, while the maximum number of substrates is 17 (Figure 12). The catalyst with the highest number of substrates with $\Delta\Delta G^{\ddagger}$ above the threshold is the same as shown in Figure 11.

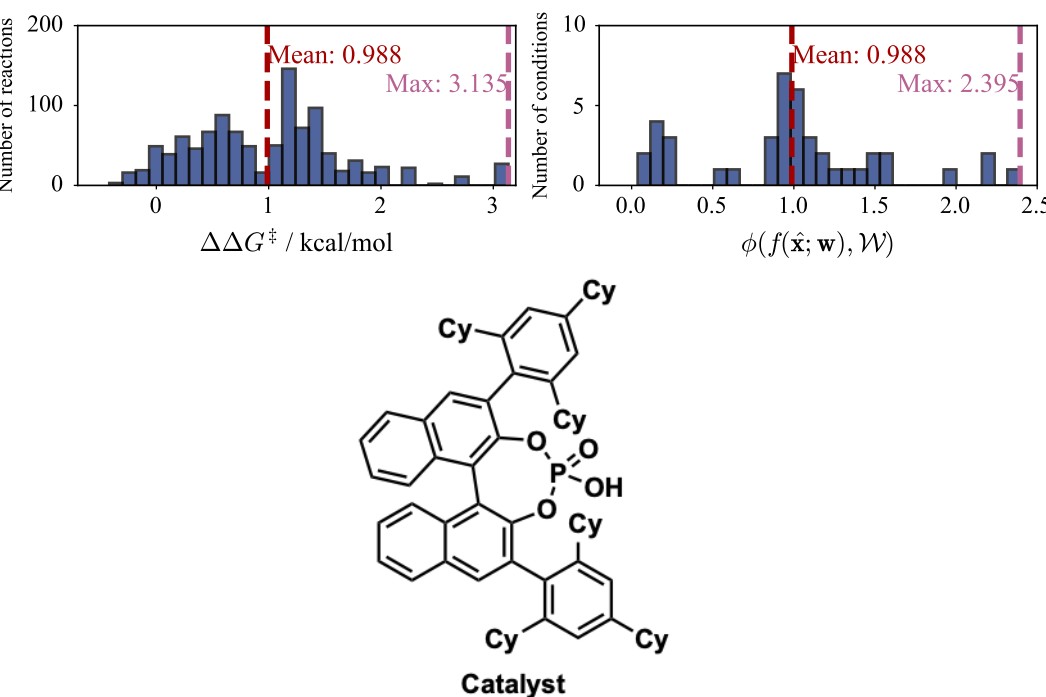

Figure 10: Reaction diagram of the N,S-Acetal formation, where an imine reacts with a thiol. Reaction conditions include a catalyst. The numbers indicate the amount of different species in the benchmark.

Figure 11: Top left: Distribution of $\Delta\Delta G^{\ddagger}$ for the N,S-Acetal formation in the original benchmark. Top right: Distribution of the average $\Delta\Delta G^{\ddagger}$ for each catalyst for the N,S-Acetal formation in the original benchmark. Bottom: Catalyst with the highest average $\Delta\Delta G^{\ddagger}$ in the original benchmark. Cy = Cyclohexyl

**Borylation reaction**

The borylation reaction benchmark is concerned with the Ni-catalyzed borylation of different aryl electrophiles (aryl chlorides, aryl bromides, and aryl sulfamates) (Figure 13). In total, 33 different aryl electrophiles were tested. As reaction conditions, ligands (23 different ligands), and solvents (2 different solvents) were varied. In total, the benchmark consists of 1518 different experiments, for which the yield is reported.

The average yield is $45.5\%$, whereas the maximum yield is $100.0\%$ (Figure 14). The average of the average yield of each condition is $45.5\%$, while the maximum of the average yield of the conditions is $65.4\%$ (Figure 14). The ligand-solvent combination with the highest average yield is shown in Figure 14.

With respect to the threshold aggregation function, the chosen threshold was $90\%$. The average number of substrates with a yield above this threshold are $1.457$, while the maximum number of substrates is 5 (Figure 15). The ligand-solvent combination with the highest number of substrates with a yield above the threshold is the same as shown in Figure 14. However, the shown ligand-solvent combination is only one of four combinations.

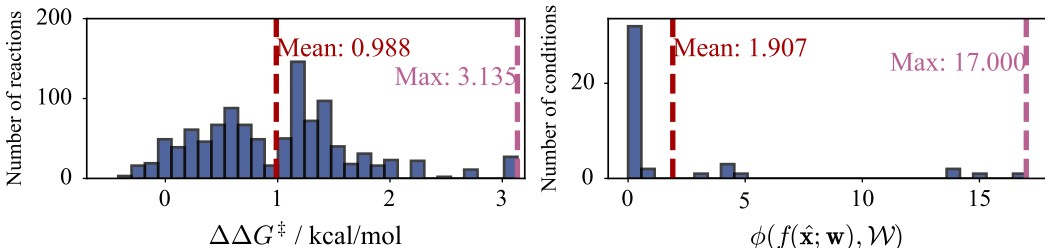

Figure 12: Left: Distribution of $\Delta\Delta G^\ddagger$ for the N,S-Acetal formation in the original benchmark. Right: Distribution number of substrates with a $\Delta\Delta G^\ddagger$ above the specified threshold for each catalyst for the N,S-Acetal formation in the original benchmark.

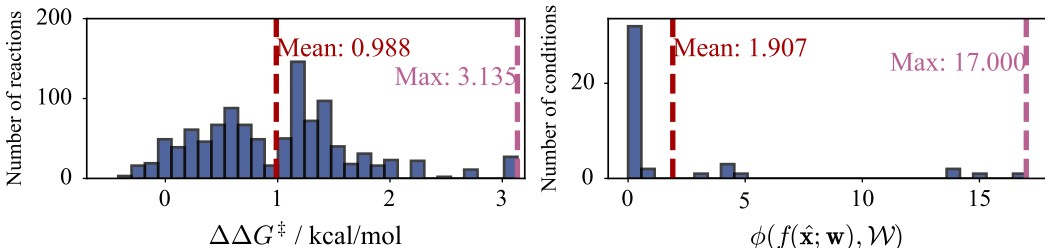

Figure 13: Reaction diagram of the borylation reaction, where different aryl electrophiles are borylated. Reaction conditions include a ligand, and a solvent. The numbers indicate the amount of different species in the benchmark.

**Deoxyfluorination reaction**

The deoxyfluorination reaction benchmark is concerned with the transformation of different alcohols into the corresponding fluorides (Figure 16). In total, 37 different alcohols were tested. As reaction conditions, sulfonyl fluorides (fluoride sources, five different fluorides) and bases (four different bases) were varied. In total, the benchmark consists of 740 different experiments, for which the yield is reported.

The average yield is $40.4\%$, whereas the maximum yield is $100.6\%$ (Figure 17). The yield larger than $100\%$ is contained in the originally published dataset. The average of the average yield of each condition is $40.4\%$, while the maximum of the average yield of the conditions is $57.2\%$ (Figure 17). The fluoride-base combination with the highest average yield is shown in Figure 17.

With respect to the threshold aggregation function, the chosen threshold was $90\%$. The average number of substrates with a yield above this threshold are $1.400$, while the maximum number of substrates is 5 (Figure 18). The fluoride-base combination with the highest number of substrates with a yield above the threshold is shown in Figure 18.

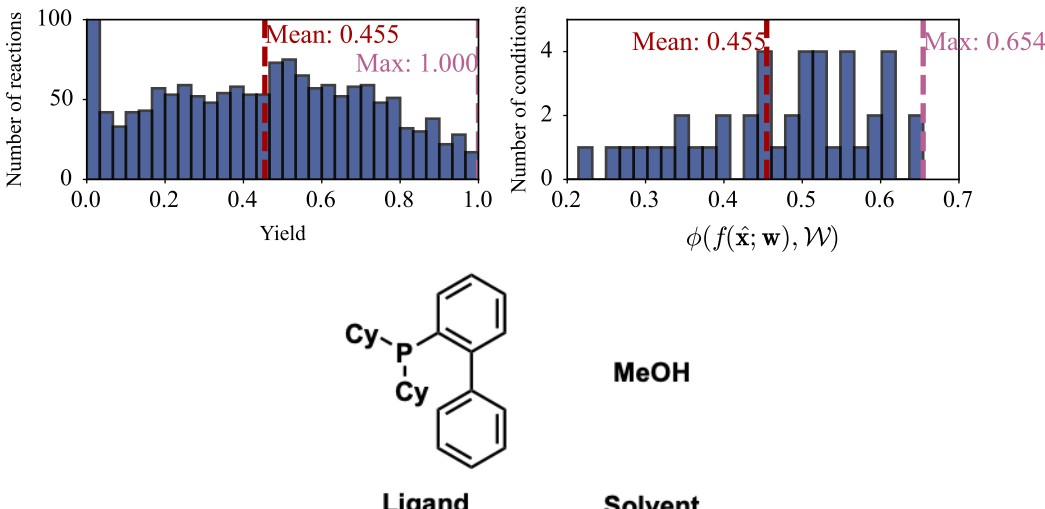

Figure 14: Top left: Distribution of the yield for the borylation reaction in the original benchmark. Top right: Distribution of the average yield for each ligand-solvent combination for the borylation reaction in the original benchmark. Bottom: Ligand-solvent combination with the highest average yield in the original benchmark. Cy = Cyclohexyl.

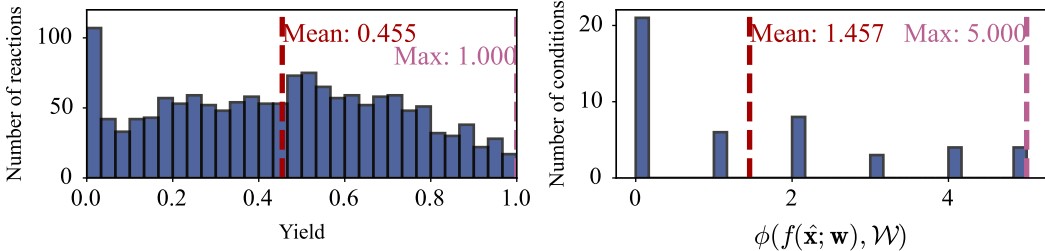

Figure 15: Left: Distribution of the yield for the borylation reaction in the original benchmark. Right: Distribution of the number of substrates with a yield above the specified threshold for each ligand-solvent combination for the borylation reaction in the original benchmark.

### A.2.2 AUGMENTATION

Since the described benchmarks consist of a high number of high-outcome experiments (the respective search spaces were rationally designed by expert chemists), we augment them with more negative examples to make them more relevant to real-world optimization campaigns. New substrates are generated by mutating the originally reported substrates via the STONED algorithm (Nigam et al., 2021). In a first filtering step, new substrates were removed if they had a Tanimoto similarity to the original substrate smaller than 0.75 (0.6 for the borylation reaction to obtain a reasonable number of additinal substrates) or if they did not possess the functional groups required for the reaction. To ensure that the benchmark is augmented with negative examples, random forests are fitted to the original benchmarks (see above). The mean absolute errors (MAEs), root mean square errors (RMSEs) and $r^2$ score ($r^2$), Spearman's rank correlation coefficient (Spearman's $\rho$) of the random forest regressors fitted to and evaluated on the original benchmarks are shown in Table 3. In addition, to evaluate the predictive utility of the random forest regressors, we perform 5-fold cross validation on the original benchmark. The MAE, RMSE, $r^2$ and Spearman's $\rho$ of the 5-fold cross validation are reported in Table 4. Even though the predictive performance on the CV does not achieve a high Spearman's rank coefficient, the comparably low MAEs and RMSEs, as well as high $r^2$ values suggest that they are a reasonable oracle. Newly generated substrates were incorporated if the average reaction outcome over all reported reaction conditions is below a defined threshold. The

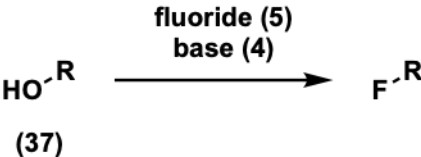

Figure 16: Reaction diagram of the deoxyfluorination reaction, where an alcohol is transformed to the corresponding fluoride. Reaction conditions include a fluoride source, and a base. The numbers indicate the amount of different species in the benchmark.

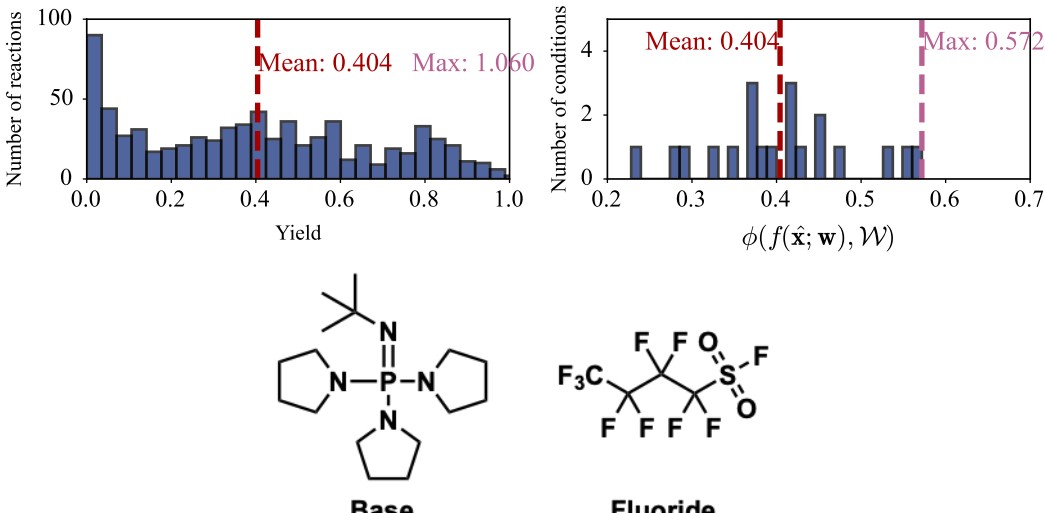

Figure 17: Top left: Distribution of the yield for the deoxyfluorination reaction in the original benchmark. Top right: Distribution of the average yield for each fluoride-base combination for the deoxyfluorination reaction in the original benchmark. Bottom: Fluoride-base combination with the highest average yield in the original benchmark.

chosen thresholds are $1.0\%$ for the Pd-catalyzed carbon-heteroatom coupling, $0.7\,\text{kcal/mol}$ for the N,S-Acetal formation, $12\%$ for the borylation reaction, and $5\%$ for the deoxyfluorination reaction. If a substrate passed these filters, the reactions with all different reported conditions were added, with reaction outcomes being taken from as predicted from the random forest emulator.

Table 3: MAE, RMSE, $r^2$, and Spearman's $\rho$ of random forest regressors fitted to and evaluated on the original benchmark problems.

| Benchmark problem | MAE | RMSE | $r^2$ | Spearman's $\rho$ |
|---|---|---|---|---|
| Pd-catalyzed coupling | $3.16 \times 10^{-3}$ | $8.75 \times 10^{-3}$ | 0.966 | 0.898 |
| N,S-Acetal formation | $4.95 \times 10^{-2}$ kcal/mol | $7.39 \times 10^{-2}$ kcal/mol | 0.989 | 0.994 |
| Borylation reaction | $3.62 \times 10^{-2}$ | $4.92 \times 10^{-2}$ | 0.966 | 0.987 |
| Deoxyfluorination | $2.13 \times 10^{-2}$ | $3.38 \times 10^{-2}$ | 0.986 | 0.993 |

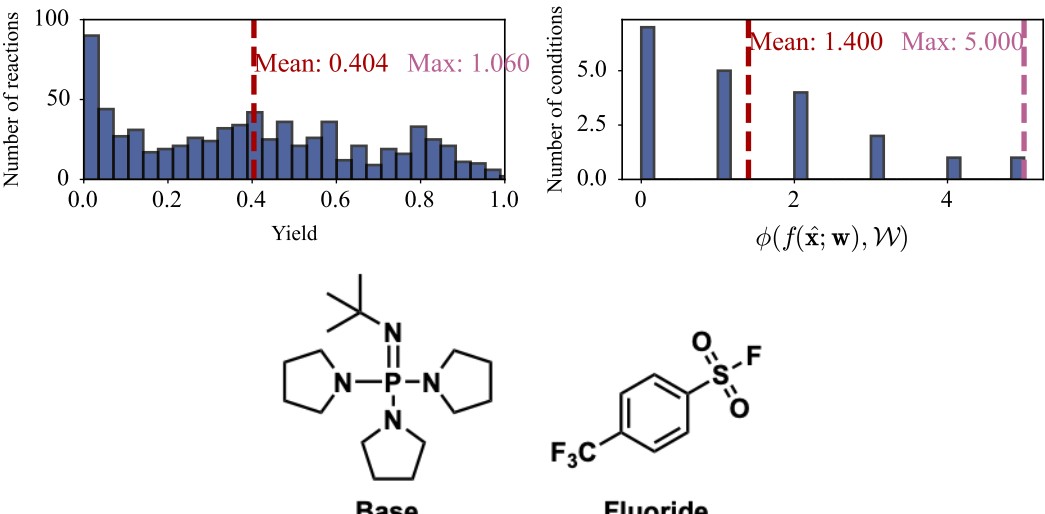

Figure 18: Top left: Distribution of the yield for the deoxyfluorination reaction in the original benchmark. Top right: Distribution of the number of substrates with a yield above the specified threshold for each fluoride-base combination for the deoxyfluorination reaction in the original benchmark. Bottom: Fluoride-base combination with the highest number of substrate with a yield above the threshold in the original benchmark.

Table 4: MAE, RMSE, r$^2$, and Spearman's rank correlation coefficient with their standard errors of random forest regressors in a 5-fold cross validation on the original benchmark problems.

| Benchmark problem | MAE | RMSE | r$^2$ | Spearman's $\rho$ |
|---|---|---|---|---|
| Pd-catalyzed coupling | $(9.3 \pm 0.7) \times 10^{-3}$ | $(2.44 \pm 0.18) \times 10^{-2}$ | $0.73 \pm 0.03$ | $0.429 \pm 0.007$ |
| N,S-Acetal formation | $(1.43 \pm 0.07) \times 10^{-1}$ kcal/mol | $(2.11 \pm 0.10) \times 10^{-1}$ kcal/mol | $0.908 \pm 0.010$ | $0.474 \pm 0.007$ |
| Borylation reaction | $(1.04 \pm 0.03) \times 10^{-1}$ | $(1.39 \pm 0.04) \times 10^{-1}$ | $0.729 \pm 0.013$ | $0.425 \pm 0.009$ |
| Deoxyfluorination | $(5.96 \pm 0.14) \times 10^{-2}$ | $(8.42 \pm 0.15) \times 10^{-2}$ | $0.913 \pm 0.004$ | $0.478 \pm 0.003$ |

### A.2.3 AUGMENTED BENCHMARK PROBLEMS

**Pd-catalyzed carbon-heteroatom coupling**

Augmentation increases the number of different nucleophiles from 16 to 31 (see Figure 19). Combined with the 96 reported reaction condition combinations, the augmented dataset consists of 2976 reactions, for which the conversion is reported.

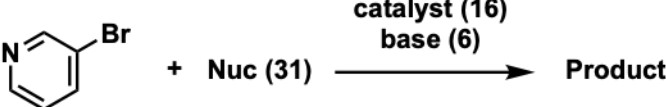

Figure 19: Reaction diagram of the Pd-catalyzed carbon-heteroatom coupling, where 3-bromopyridine reacts with a nucleophile. Reaction conditions include a catalyst and a base. The numbers indicate the amount of different species in the augmented benchmark.

Augmentation decreased the average conversion from 2.05% to 1.34%, whereas the maximum conversion remained the same at 39.81% (see Figure 20). The average of the average conversion of each

condition is decreased from $2.05\%$ to $1.34\%$, and the maximum of the average conversion of each condition is also decreased from $7.60\%$ to $6.00\%$ (see Figure 20). The catalyst-base combination with the highest average conversion is unaffected by the augmentation and shown in Figure 20.

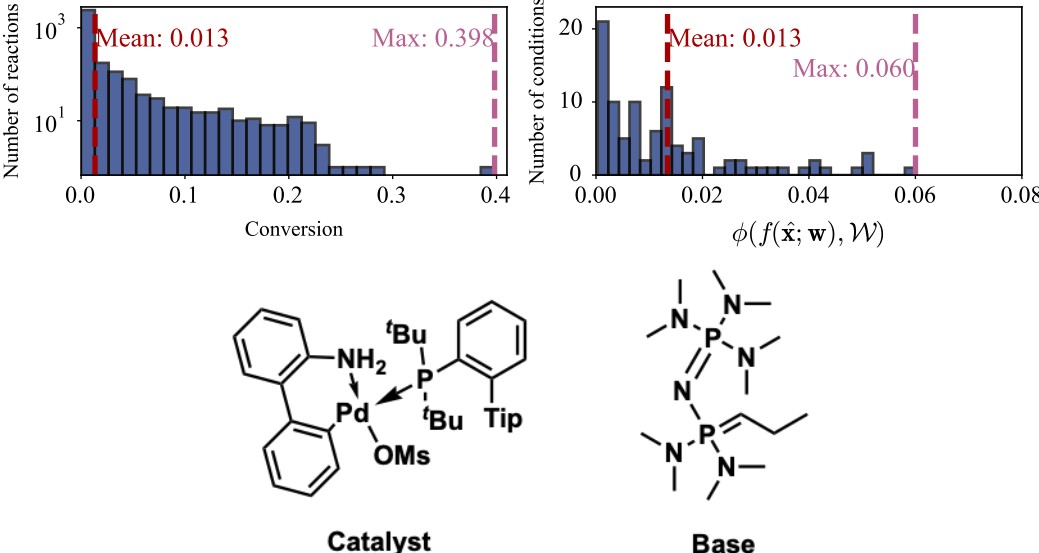

Figure 20: Top left: Distribution of the conversion for the Pd-catalyzed carbon-heteroatom coupling in the augmented benchmark. Top right: Distribution of the average conversion for each catalyst-base combination for the Pd-catalyzed carbon-heteroatom coupling in the augmented benchmark. Bottom: Catalyst-base combination with the highest average conversion in the augmented benchmark. Tip = 2,4,6-triisopropylphenyl.

With respect to the threshold aggregation function, the chosen threshold was $7.50\%$. The average number of substrates with a conversion above this threshold are $1.646$, while the maximum number of substrates is 8 (Figure 21). The catalyst-base combination with the highest number of substrates with a conversion above the threshold is the same as shown in Figure 20.

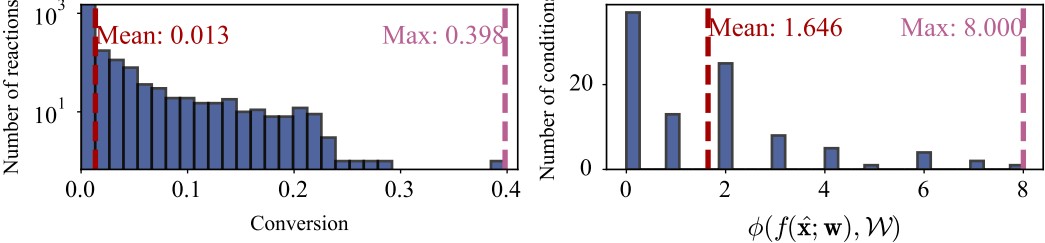

Figure 21: Left: Distribution of the conversion for the Pd-catalyzed carbon-heteroatom coupling in the augmented benchmark. Right: Distribution of the number of substrates with a conversion above the specified threshold for each catalyst-base combination for the Pd-catalyzed carbon-heteroatom coupling in the augmented benchmark.

**N,S-Acetal formation**

Augmentation increases the number of thiols from five to 13, while the number of imines remained constant at five (see Figure 22). Combined with the 43 reported reaction conditions, the augmented benchmark consists of 2795 reactions, for which $\Delta\Delta G^{\ddagger}$ is reported.

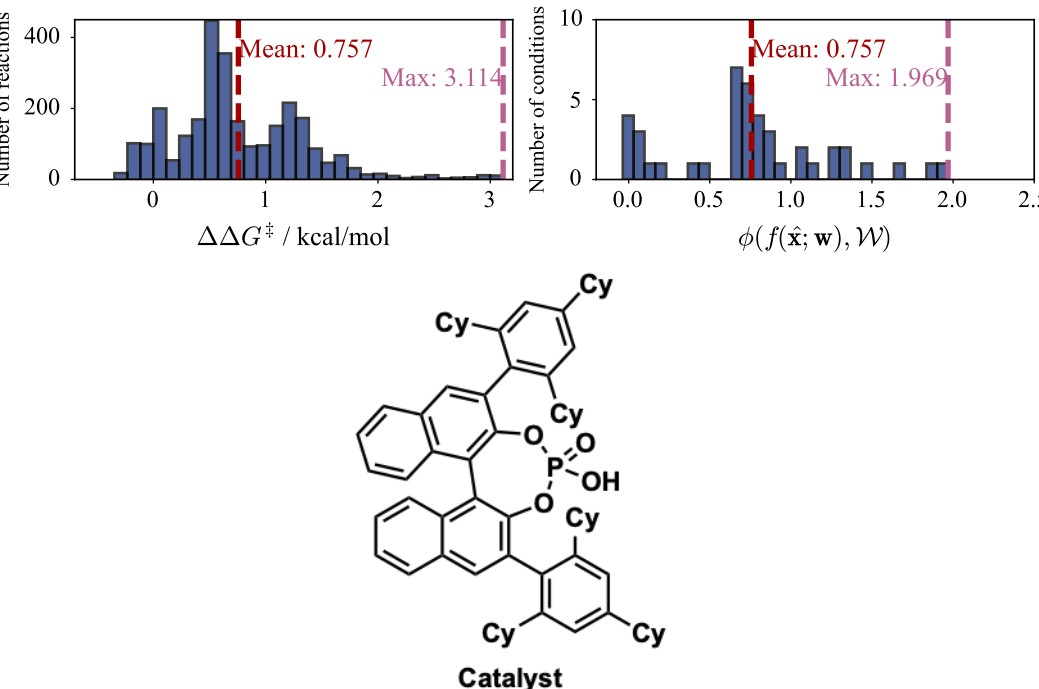

Figure 22: Reaction diagram of the N,S-Acetal formation, where an imine reacts with a thiol. Reaction conditions include a catalyst. The numbers indicate the amount of different species in the augmented benchmark.

Augmentation decreased the average $\Delta\Delta G^{\ddagger}$ from $0.988$ kcal/mol to $0.757$ kcal/mol, whereas the maximum $\Delta\Delta G^{\ddagger}$ was slightly decreased from $3.135$ kcal/mol to $3.114$ kcal/mol (see Figure 23). This decrease is due to the fact that the augmented benchmark only contains values are taken as predicted by the random forest emulator (to investigate optimization performance, the random forest emulator is taken for both the original and augmented benchmarks). Through augmentation, the average of the average $\Delta\Delta G^{\ddagger}$ of each condition decreased from $0.988$ kcal/mol to $0.757$ kcal/mol, while the maximum of the average $\Delta\Delta G^{\ddagger}$ of all conditions decreased as well from $2.395$ kcal/mol to $1.969$ kcal/mol (see Figure 23). The catalyst with the highest average $\Delta\Delta G^{\ddagger}$ is unaffected by the augmentation and shown in Figure 23.

Figure 23: Top left: Distribution of $\Delta\Delta G^{\ddagger}$ for the N,S-Acetal formation in the augmented benchmark. Top right: Distribution of the average $\Delta\Delta G^{\ddagger}$ for each catalyst for the N,S-Acetal formation in the augmented benchmark. Bottom: Catalyst with the highest average $\Delta\Delta G^{\ddagger}$ in the augmented benchmark. Cy = Cyclohexyl.

With respect to the threshold aggregation function, the chosen threshold was $2.0$ kcal/mol. The average number of substrates with $\Delta\Delta G^{\ddagger}$ above this threshold are $1.814$, while the maximum number of

substrates is 16 (Figure 24). The catalyst with the highest number of substrates with $\Delta\Delta G^{\ddagger}$ above the threshold is the same as shown in Figure 23.

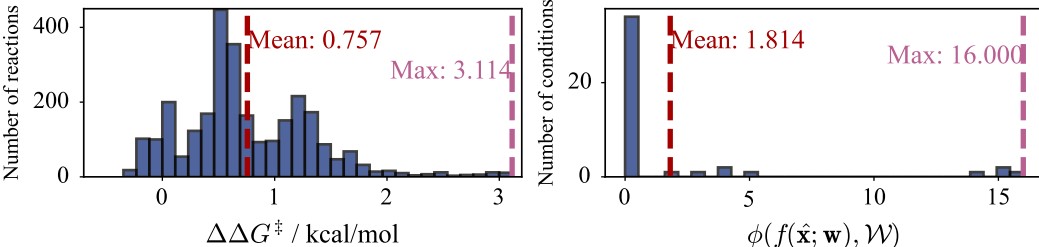

Figure 24: Left: Distribution of $\Delta\Delta G^{\ddagger}$ for the N,S-Acetal formation in the augmented benchmark. Right: Distribution number of substrates with a $\Delta\Delta G^{\ddagger}$ above the specified threshold for each catalyst for the N,S-Acetal formation in the augmented benchmark.

**Borylation reaction**

Augmentation increases the number of different aryl electrophiles from 33 to 75 (see Figure 25). Combined with the 46 reported reaction condition combinations, the augmented dataset consists of 3450 reactions, for which the yield is reported.

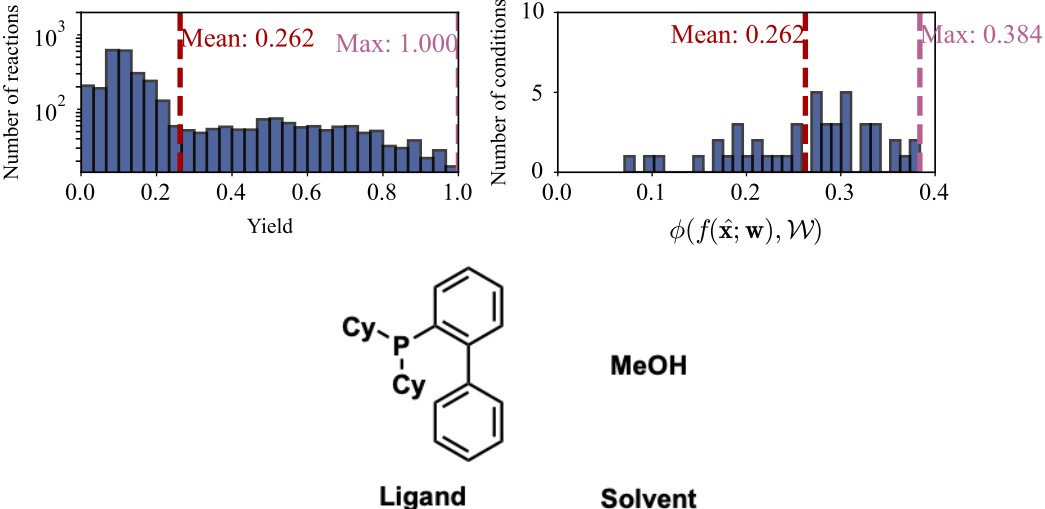

Figure 25: Reaction diagram of the borylation reaction, where an aryl electrophile is borylated via a nickel catalyst. Reaction conditions include a ligand, and a solvent. The numbers indicate the amount of different species in the augmented benchmark.

Augmentation decreased the average yield from $45.5\%$ to $26.2\%$, whereas the maximum yield remained the same at $100.0\%$ (see Figure 26). The average of the average yield of each condition is decreased from $45.5\%$ to $26.2\%$, and the maximum of the average yield of each condition is also decreased from $65.4\%$ to $38.4\%$ (see Figure 26). The ligand-solvent combination with the highest average yield is unaffected by dataset and augmentation and shown in Figure 26.

Figure 26: Top left: Distribution of the yield for the borylation reaction in the augmented benchmark. Top right: Distribution of the average yield for each ligand-solvent combination for the borylation reaction in the augmented benchmark. Bottom: Ligand-solvent combination with the highest average yield in the augmented benchmark. Cy = Cyclohexyl.

With respect to the threshold aggregation function, the chosen threshold was $90\%$. The average number of substrates with a yield above this threshold are $1.457$, while the maximum number of substrates is $5$ (Figure 27). Several ligand-solvent combinations provide the highest number of substrates with a yield above the threshold, one of them is shown in Figure 26. The ligand-solvent combinations are unaffected by the augmentation.

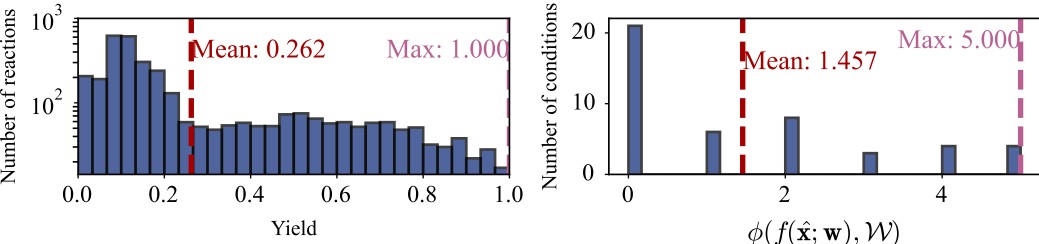

Figure 27: Left: Distribution of the yield for the borylation reaction in the augmented benchmark. Right: Distribution of the number of substrates with a yield above the specified threshold for each ligand-solvent combination for the borylation reaction in the augmented benchmark.

**Deoxyfluorination reaction**

Augmentation increases the number of different alcohols from 37 to 54 (see Figure 28). Combined with the 20 reported reaction condition combinations, the augmented dataset consists of 1080 reactions, for which the yield is reported.

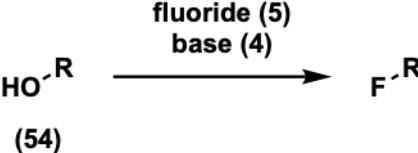

Figure 28: Reaction diagram of the deoxyfluorination reaction, where an alcohol is converted to the corresponding fluoride. Reaction conditions include a fluoride source and a base. The numbers indicate the amount of different species in the augmented benchmark.

Augmentation decreased the average yield from $40.4\%$ to $28.9\%$, whereas the maximum yield remained the same at $100.6\%$ (see Figure 29). The yield larger than $100\%$ is contained in the originally published dataset. The average of the average yield of each condition is decreased from $40.4\%$ to $28.9\%$, and the maximum of the average yield of each condition is also decreased from $57.2\%$ to $43.8\%$ (see Figure 29). The fluoride-base combination with the highest average yield is unaffected by augmentation and shown in Figure 29.

With respect to the threshold aggregation function, the chosen threshold was $90\%$. The average number of substrates with a yield above this threshold are $1.400$, while the maximum number of substrates is 5 (Figure 30). The fluoride-base combination with the highest number of substrates with a yield above the threshold is also unaffected by augmentation and shown in Figure 30.

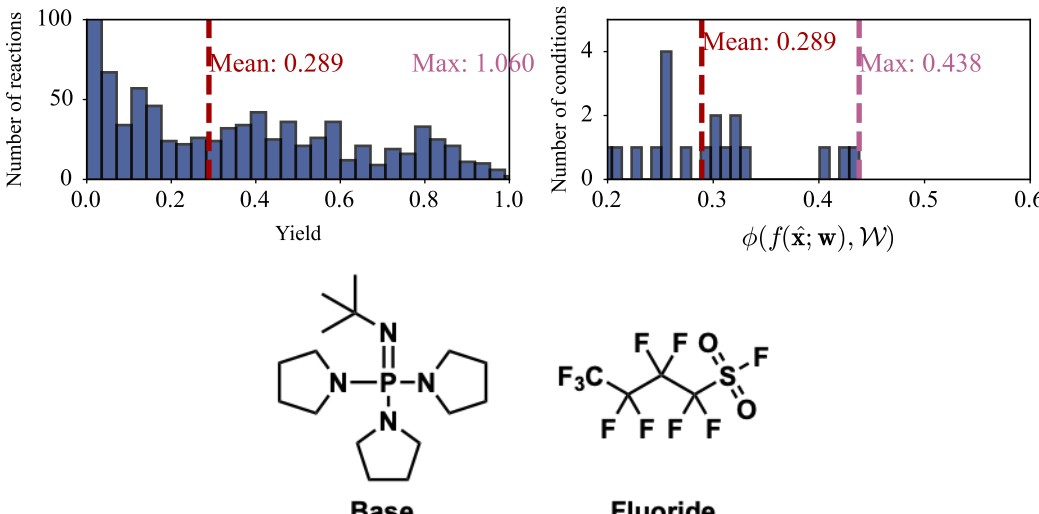

Figure 29: Top left: Distribution of the yield for the deoxyfluorination reaction in the augmented benchmark. Top right: Distribution of the average yield for each fluoride-base combination for the deoxyfluorination reaction in the augmented benchmark. Bottom: Fluoride-base combination with the highest average yield in the augmented benchmark.

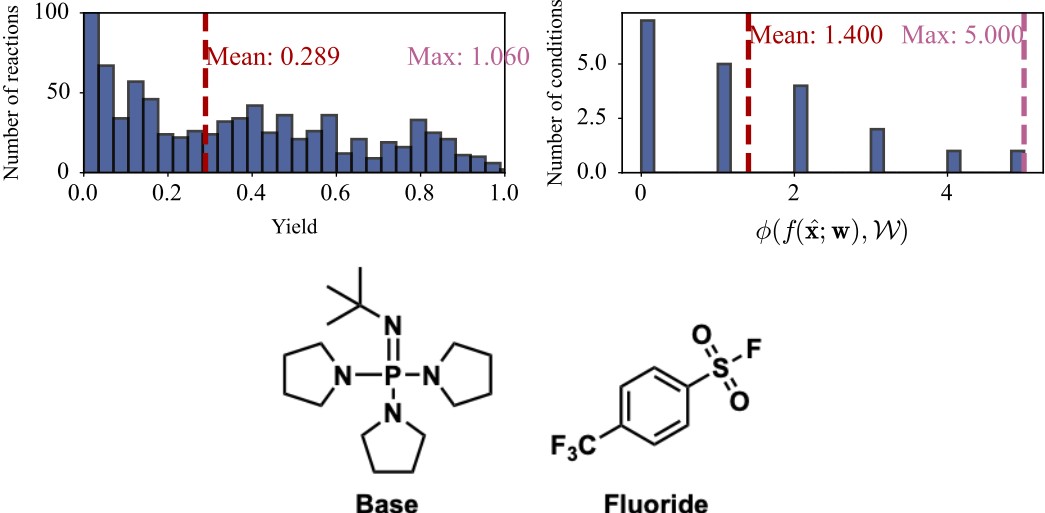

Figure 30: Top left: Distribution of the yield for the deoxyfluorination reaction in the augmented benchmark. Top right: Distribution of the number of substrates with a yield above the specified threshold for each fluoride-base combination for the deoxyfluorination reaction in the augmented benchmark. Bottom: Fluoride-base combination with the highest number of substrate with a yield above the threshold in the augmented benchmark.

### A.3 Grid Search for Analyzing Benchmark Problems

To analyse the utility of considering multiple substrates in an optimization campaign, we performed exhaustive grid search on the described benchmark problems. For each problem, the substrates were split into an initial train and test set among the substrates. In total, thirty different train/test splits were performed. The obtained train set was further subsampled into smaller training sets with varying sizes to investigate the influence on the number of substrates. Sampling among the substrates in the train set was performed either through random sampling, farthest point sampling or "Average Sampling", where the required number of substrates was chosen as the substrates with

the highest average Tanimoto similarity to all other train substrates. For each subsampled training set, the most general conditions were identified via exhaustive grid search. The general reaction outcome, as specified by the aggregation function, is evaluated for these conditions on the held-out test set. Further, this general reaction outcome was scaled from 0 to 1 to give a dataset independent generality score, where 0 is the worst possible general reaction outcome for the given test set and 1 is the best possible general reaction outcome for the test set. Hence, this score should be maximized. For the different benchmark problems, we report this generality score, where we also compare the behaviour of the original and augmented problems. Below, the results of the described data analysis are shown for the benchmark problems not shown in the main text.

### A.4 DETAILS ON BO FOR GENERALITY BENCHMARKING

To identify whether BO for generality, as described above, can efficiently identify the general optima, we conducted several benchmarking runs on the described benchmark problems. On each problem, we perform benchmarking for multiple optimization strategies, as listed in Table 2.

In each optimization campaign, we used a single-task GP regressor, as implemented in *GPyTorch* (Gardner et al., 2018), with a TanimotoKernel as implemented in *Gauche* (Griffiths et al., 2023). Molecules were represented using Morgan Fingerprints (Morgan, 1965) with 1024 bits and a radius of 2. Fingerprints were generated using RDKit (Landrum, 2023). It is notable that, while such a representation was chosen due to its suitability for broad chemical spaces, more specific representations such as descriptors might be able to improve the optimization performance.

The acquisition policies were benchmarked on all benchmark problems with differently sampled substrates for each optimization run. For each benchmark, we selected the train set randomly, consisting of twelve nucleophiles in the Pd-catalyzed carbon-heteroatom coupling benchmark, three imines and three thiols in the N,S-Acetal formation benchmark, twentyfive alcohols in the Deoxyfluorination reaction, and twenty aryl halides in the Borylation reaction. Thirty independent optimization campaigns were performed for each. The generality of the proposed general conditions at each step during the optimization is shown.

### A.5 DETAILS ON BANDIT ALGORITHM BENCHMARKING

The benchmarking of BANDIT (Wang et al., 2024) was performed across the benchmark problems using their proposed UCB1TUNED algorithm with differently sampled substrates for the optimization. For each benchmark, we selected the train set randomly, consisting of twelve nucleophiles in the Pd-catalyzed carbon-heteroatom coupling benchmark, three imines and three thiols in the N,S-Acetal formation benchmark, twentyfive alcohols in the Deoxyfluorination reaction, and twenty aryl halides in the Borylation reaction. Thirty independent optimization campaigns were performed for each. To ensure fair comparison, the ground truth was set to be the proxy function calculated for each dataset. To select the optimum $\mathbf{x}$ value at each step $k$, we relied on the authors definition of the best arm as the most sampled arm at step $k$.

### A.6 ADDITIONAL RESULTS AND DISCUSSION

#### A.6.1 ADDITIONAL RESULTS ON THE DATASET ANALYSIS FOR UTILITY OF GENERALITY-ORIENTED OPTIMIZATION

In addition to analysing the utility of generality-oriented optimization for $\phi$ as the mean aggregation, which is shown in Figure 3, we also perform a similar analysis for $\phi$ as the threshold aggregation, where the chosen thresholds are as described in Appendix A.2. The results of this analysis are shown in Figure 31. Similar to the case where $\phi$ is the mean aggregation, we observe that in the majority of benchmark problems, more general reaction conditions are obtained by considering multiple substrates. The only exemption to this observation is the Deoxyfluorination reaction benchmark, a benchmark with a particularly low number of conditions with a high threshold aggregation value (see Figure 30). In addition, we also observe a highly similar behaviour of the original and augmented benchmarks, which is due to the addition of low-performing reactions in the augmentation, which only slightly influences the results of the threshold (i.e. number of high-performing reactions).

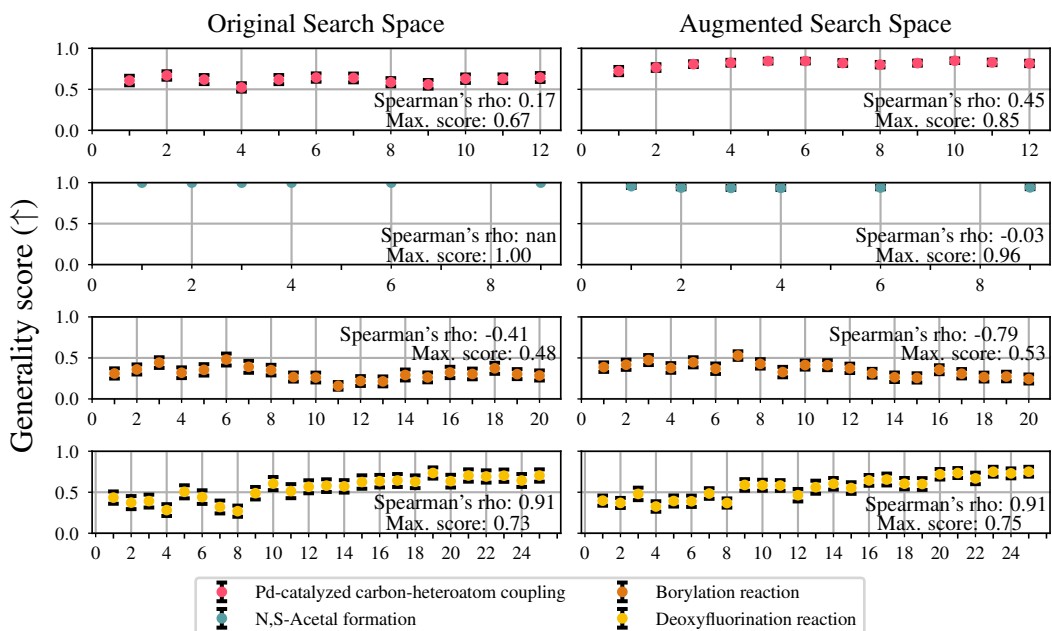

Figure 31: Normalized test-set generality score as determined by exhaustive grid search for the four benchmarks on the original (left) and augmented (right) problems for the threshold aggregation. Average and standard error are taken from thirty different train/test substrates splits.

Furthermore, we studied how different sampling techniques among the train set substrates influence the obtained generality scores. As sampling techniques, we used random sampling, farthest point sampling and "average sampling", as outlined in Appendix A.3. For $\phi$ as the mean aggregation, the results for the four different benchmarks are shown in Figure 32, Figure 33, Figure 34, and Figure 35. For $\phi$ as the threshold aggregation, the results for the two different benchmarks are shown in Figure 36, Figure 37, Figure 38, and Figure 39. Throughout the different benchmarks and aggregation functions, we observe that the generality score obtained through using the sampled train substrates are highly similar and no method clearly outperforms the others. It is particularly notable that farthest point sampling did not outperform other sampling techniques, as this strategy is commonly used to select chemicals to broadly cover chemical space (Henle et al., 2020; Gensch et al., 2022a;b; Schnitzer et al., 2024). We hypothesize that this method insensitivity is due to the low number of substrates chosen for the train set, which was chosen to still reflect realistic experimental cases.

### A.6.2 ADDITIONAL RESULTS ON THE BENCHMARKING ON THE AUGMENTED BENCHMARKS

In addition to the experiments shown in the main text, we benchmarked the sequential one-step and two-step lookahead functions where either a single substrate is selected or in the complete monitoring case. For both the one-step and two-step lookahead acquisition strategies we observe a significant loss in optimization efficiency for generality-oriented optimization, when only a single substrate is considered (see Figure 40). This is expected, as the constant observation of only one substrate does provide limited information into how different substrates might react, which is unsuitable for generality-oriented optimization. Similarly, the results shown in Figure 41 clearly demonstrate that a complete monitoring scenario is not optimally efficient for generality-oriented optimization. We hypothesize that this is because the $\mathcal{X}$ can be more efficiently explored, as not every substrate has to be tested for a specific set of reaction conditions. This underlines the utility of improved and efficient decision-making algorithms in complex optimization scenarios.

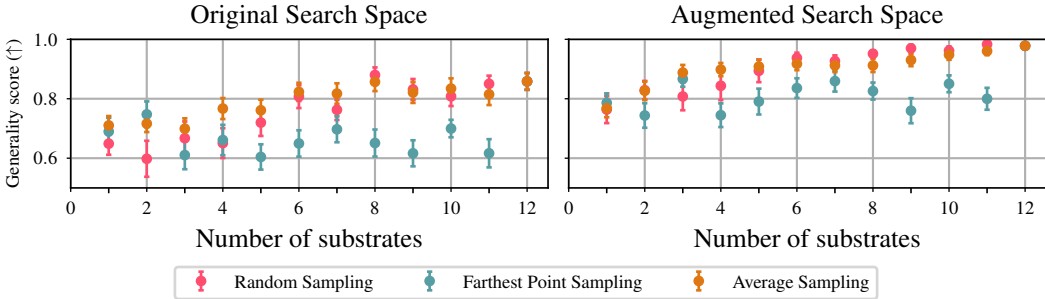

Figure 32: Generality score as determined by exhaustive grid search for the Pd-catalyzed carbon-heteroatom coupling benchmark on the original (left) and augmented (right) problems for the mean aggregation as $\phi$. Average and standard error are taken from thirty different train/test substrates splits.

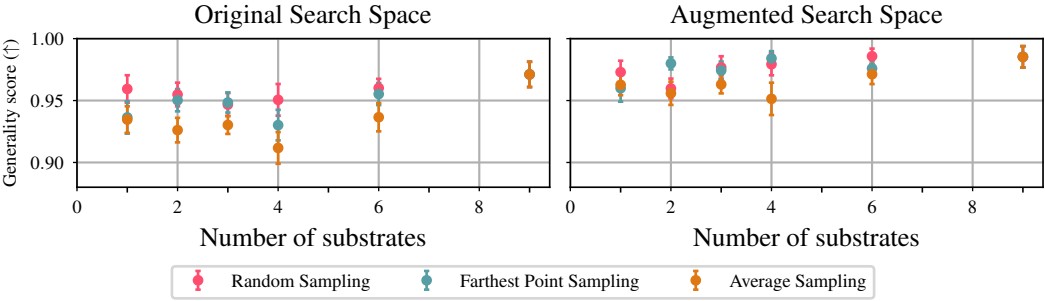

Figure 33: Generality score as determined by exhaustive grid search for the N,S-Acetal formation benchmark on the original (left) and augmented (right) problems for the mean aggregation as $\phi$. Average and standard error are taken from thirty different train/test substrates splits.

### A.6.3 ADDITIONAL RESULTS ON THE BENCHMARKING ON THE ORIGINAL BENCHMARKS

In addition to the results described above, we also benchmark the strategies described in Table 2 on the original benchmarks. In general, we observe highly similar results compared to the augmented benchmarks that have already been discussed. This emphasizes that, while augmentation of established benchmarks remains necessary to reflect real-world conditions, the conclusions on algorithmic performances remain largely unaffected from the biases within the benchmarks. A high robustness in optimization performance on benchmark distribution further increases the utility of generality-oriented optimization in the laboratory.

Specifically, we find that, similar to the augmented benchmarks, the SEQ 1LA-UCB-PV strategy shows a significantly better optimization performance than other algorithms published in the chemical domain (see Figure 42). Comparing multiple one-step and two-step lookahead acquisition strategies, with varying $\alpha_x$ again emphasizes that both strategies perform similarly and that an explorative acquisition of $\mathbf{x}_{\text{next}}$ is crucial for successful generality-oriented optimization (see Figure 43). Confirming results from the augmented benchmarks, we also observe that a variation in $\alpha_w$ does not affect the optimization performance of the one-step lookahead acquisition strategy, while a random acquisition of $\mathbf{w}_{\text{next}}$ leads to less efficient optimizations for two-step lookahead strategies (see Figure 43). In addition, we also confirm the surprising empirical observation that a joint acquisition of $\mathbf{x}_{\text{next}}$ and $\mathbf{w}_{\text{next}}$ does not yield to a significantly improved optimization performance compared to a sequential acquisition (see Figure 44).

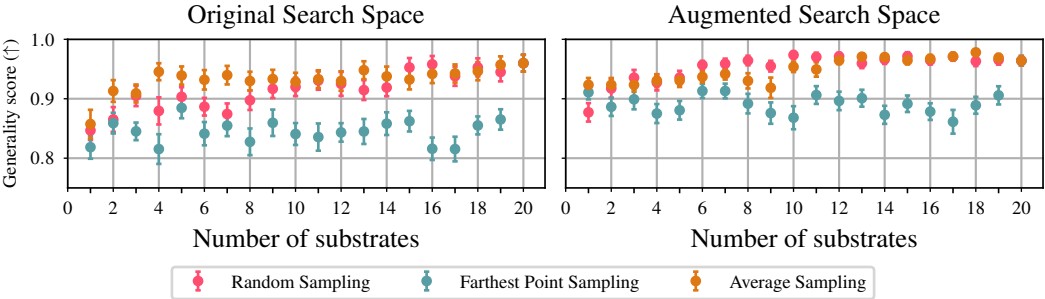

Figure 34: Generality score as determined by exhaustive grid search for the Borylation reaction benchmark on the original (left) and augmented (right) problems for the mean aggregation as $\phi$. Average and standard error are taken from thirty different train/test substrates splits.

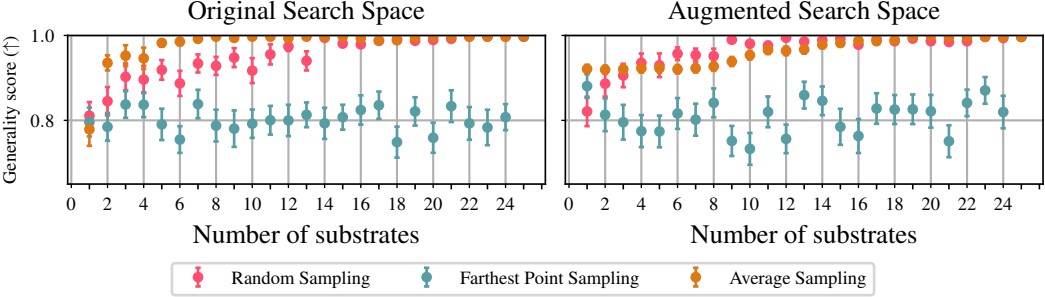

Figure 35: Generality score as determined by exhaustive grid search for the Deoxyfluorination reaction benchmark on the original (left) and augmented (right) problems for the mean aggregation as $\phi$. Average and standard error are taken from thirty different train/test substrates splits.

Lastly, we also demonstrate that a generality-oriented optimization with a single substrate and in the complete monitoring case leads to suboptimal optimization performance, as shown in Figure 45 and Figure 46, respectively.

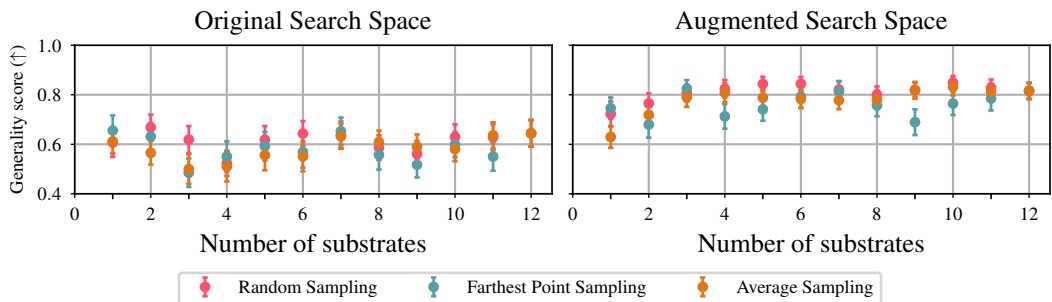

Figure 36: Generality score as determined by exhaustive grid search for the Pd-catalyzed carbon-heteroatom coupling benchmark on the original (left) and augmented (right) problems for the threshold aggregation as $\phi$. Average and standard error are taken from thirty different train/test substrates splits.

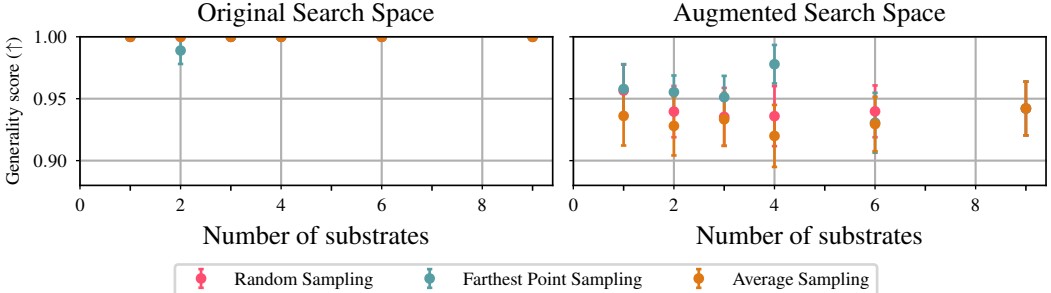

Figure 37: Generality score as determined by exhaustive grid search for the N,S-Acetal formation benchmark on the original (left) and augmented (right) problems for the threshold aggregation as $\phi$. Average and standard error are taken from thirty different train/test substrates splits.

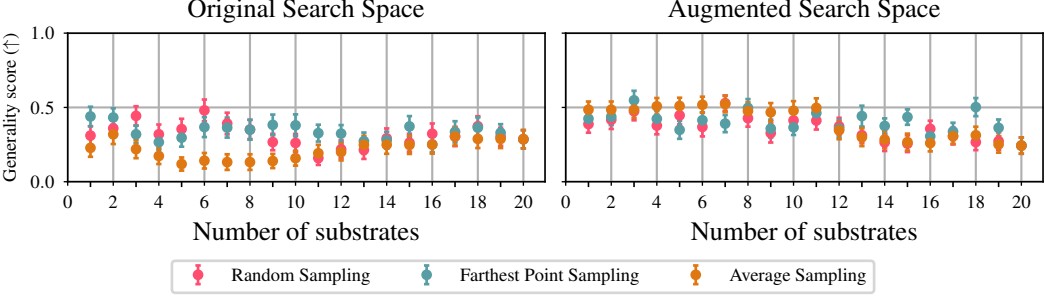

Figure 38: Generality score as determined by exhaustive grid search for the Borylation reaction benchmark on the original (left) and augmented (right) problems for the threshold aggregation as $\phi$. Average and standard error are taken from thirty different train/test substrates splits.

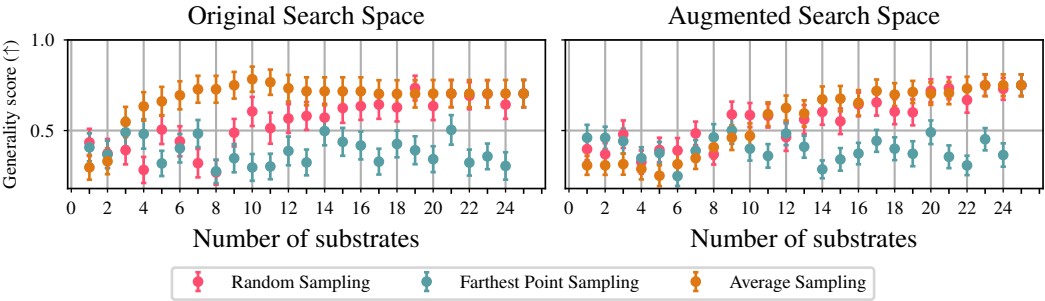

Figure 39: Generality score as determined by exhaustive grid search for the Deoxyfluorination reaction benchmark on the original (left) and augmented (right) problems for the threshold aggregation as $\phi$. Average and standard error are taken from thirty different train/test substrates splits.

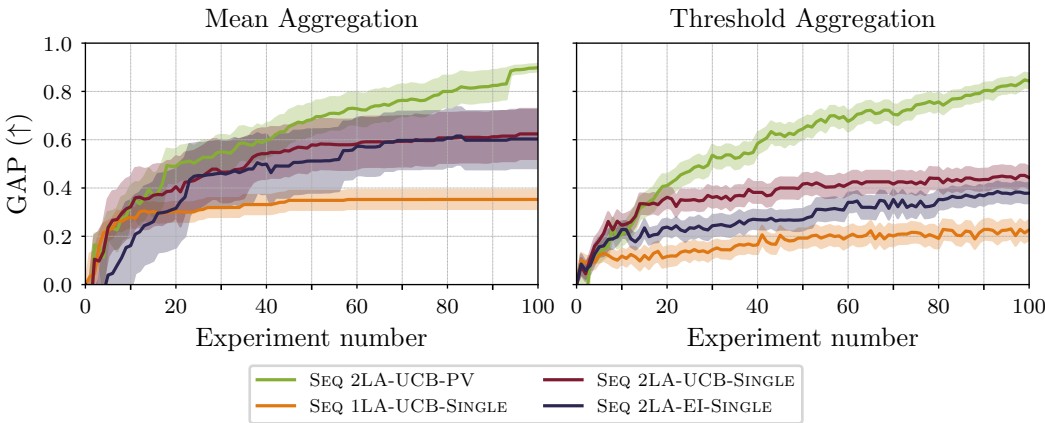

Figure 40: Optimization trajectories of different algorithms used for generality-oriented optimization considering multiple or a single substrate. The trajectories are averaged over all augmented benchmark problems with the mean (left) and threshold (right) aggregations. Optimization algorithms are described in Table 1.

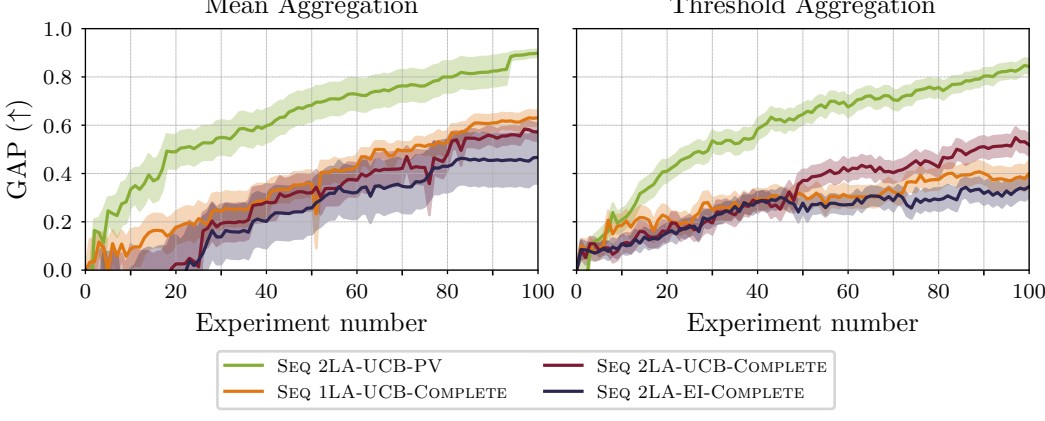

Figure 41: Optimization trajectories of different algorithms used for generality-oriented optimization considering the partial or complete monitoring case, respectively. The trajectories are averaged over all augmented benchmark problems with the mean (left) and threshold (right) aggregations. Optimization algorithms are described in Table 1.

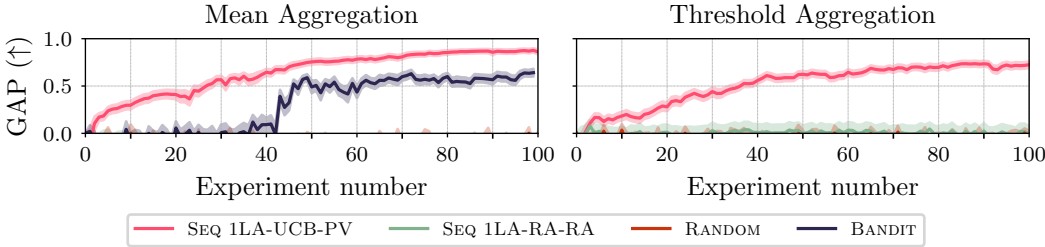

Figure 42: Optimization trajectories of different algorithms used for generality-oriented optimization in the chemical domain. The trajectories are averaged over all original benchmark problems with the mean (left) and threshold (right) aggregations. Optimization algorithms are described in Table 1.

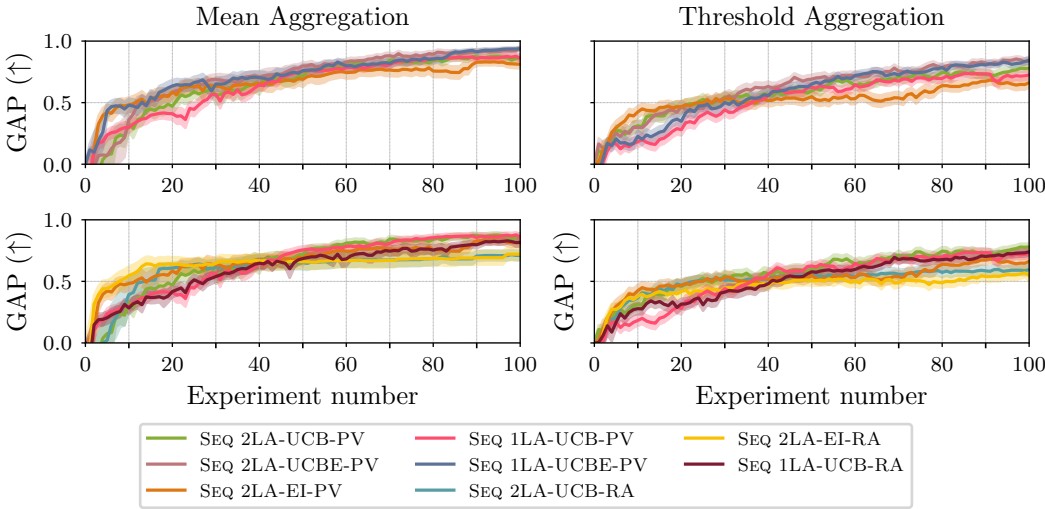

Figure 43: Optimization trajectories of different sequential acquisition strategies for generality-oriented optimization. The top row shows the influence of variation of the acquisition strategy of $\mathbf{x}_{\text{next}}$ (i.e., variation of $\alpha_x$), while the bottom row shows the influence of variation of the acquisition strategy of $\mathbf{w}_{\text{next}}$ (i.e., variation of $\alpha_w$). The trajectories are averaged over all original benchmark problems with the mean (left) and threshold (right) aggregations. Optimization algorithms are described in Table 1.

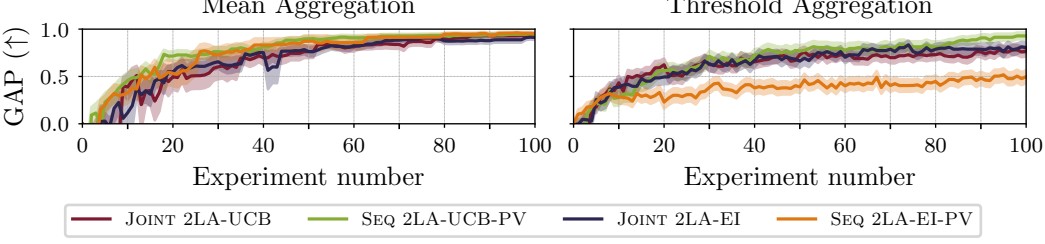

Figure 44: Optimization trajectories of sequential and joint two-step lookahead acquisition strategies for generality-oriented optimization. The trajectories are averaged over the N,S-Acetal formation and Deoxyfluorination reaction original benchmark problems with the mean (left) and threshold (right) aggregations. Optimization algorithms are described in Table 1.

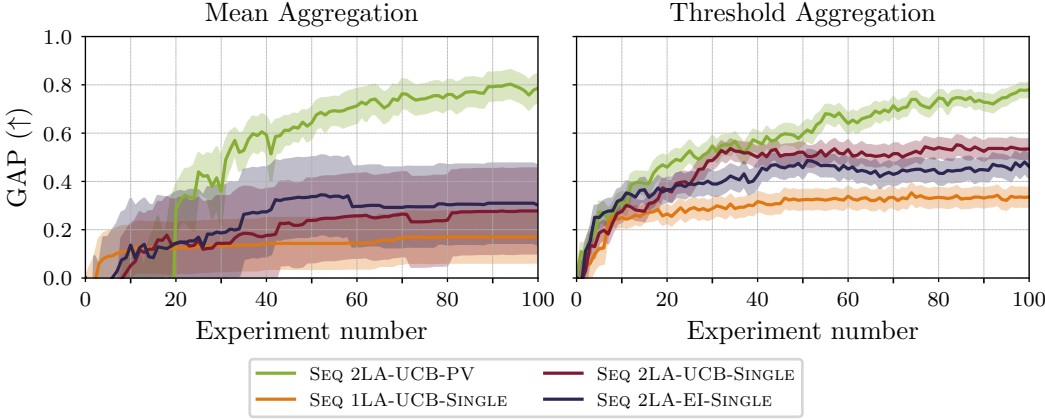

Figure 45: Optimization trajectories of different algorithms used for generality-oriented optimization considering multiple or a single substrate. The trajectories are averaged over all original benchmark problems with the mean (left) and threshold (right) aggregations. Optimization algorithms are described in Table 1.

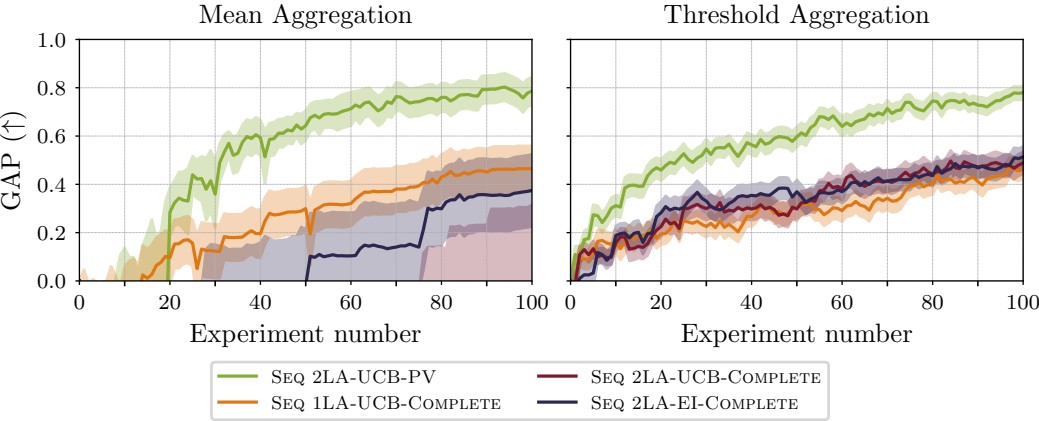

Figure 46: Optimization trajectories of different algorithms used for generality-oriented optimization considering the partial or complete monitoring case, respectively. The trajectories are averaged over all original benchmark problems with the mean (left) and threshold (right) aggregations. Optimization algorithms are described in Table 1.

