# OpenReview forum: "If Optimizing for general parameters in chemistry is useful, why is it hardly done?"
_ICLR.cc/2025/Conference — Submitted to ICLR 2025_

### Official Review · Reviewer_YKK9 · 2024-10-30

**Soundness:** 3
**Presentation:** 2
**Contribution:** 2
**Rating:** 5
**Confidence:** 3

**Summary:**

This is an applications paper where the authors look at the problem of choosing conditions for running chemical reactions for different reactants (substrates).  This is viewed as an optimisation problem where we want to maximise the expected performance based on limited data.  The task is to choose which experiments to run in order to make good predictions.  The authors test a number of different strategies, and find a random greedy algorithm does as well if not better than more sophisticated methods.

**Strengths:**

The experiments set out seem to be comprehensive and well done.  I appreciate that this presents a "negative result".  It is refreshing for authors to report results where a surprisingly simple method beats more complicated approaches.

**Weaknesses:**

I question whether this work is of high interest to researchers interested in learning representation.  There is undoubtedly value and interest in this work, but it doesn't seem to match the ICLR audience.  There is very little on representation learning.  I slightly struggled to understand this paper.  Setting up the problem in terms of generality-oriented Bayesian Optimisation seems unnecessary complicated.

There are a few sentences that did not make much sense to me.  E.g. the penultimate sentence on page 1 "Attempts to reduce..." is hard to understand.  Clearly this is a minor weakness that can be easily rectified.

**Questions:**

Is there are cleaner description of your problem as generality-oriented BO is hard to understand?
Likewise the use of currying functions makes your objectives obscure.  Are you not just trying to estimate the expected performance of some reaction conditions?

---

> ### Author Response · Authors · 2024-11-25
>
> > I question whether this work is of high interest to researchers interested in learning representation. There is undoubtedly value and interest in this work, but it doesn't seem to match the ICLR audience. There is very little on representation learning. I slightly struggled to understand this paper. Setting up the problem in terms of generality-oriented Bayesian Optimisation seems unnecessary complicated.
>
> We thank the reviewer for their perspective and the insightful comments! In line with recommendations from other reviewers, we have significantly expanded the algorithmic discussion in Section 2, making the work more accessible for an ICLR audience. Moreover, we have significantly improved the placement in the context of related BO problems, particularly discussing the connections with stochastic BO and distributionally robust BO, further clarifying the context of our problem.
>
> Combined with the largely extended scope of experiments, and the increased focus on providing a benchmark study (rather than methodological novelty), we are confident that the publication is of high interest to the ML community at ICLR. Eventually, we want to refer to the Call for Papers, which explicitly includes “applications to physical sciences (physics, chemistry, biology, etc.)” as a relevant topic for publication at this conference.
>
> > There are a few sentences that did not make much sense to me. E.g. the penultimate sentence on page 1 "Attempts to reduce..." is hard to understand. Clearly this is a minor weakness that can be easily rectified.
>
> Thank you for pointing this out. We have reformulated this particular sentence and made language clearer throughout the paper.
>
> > Is there are cleaner description of your problem as generality-oriented BO is hard to understand? Likewise the use of currying functions makes your objectives obscure. Are you not just trying to estimate the expected performance of some reaction conditions?
>
> As we have now clarified in the paper, we formulate this problem as BO over curried functions to maintain a high degree of flexibility in the problem description. This flexibility allows real-world experimentalists (chemists) to provide an arbitrary definition of generality, which can be integrated into this framework via the aggregation function. While for certain aggregation functions, BO over curried functions is known to the BO community (see Section 2.2.4), arbitrary new aggregation functions can be integrated with this problem formulation.

---

> > ### Comment · Reviewer_YKK9 · 2024-11-26
> > **Reviewers response**
> >
> > Thank you for your response.  I think concentrating more on the machine learning aspects will improve the document for the ICLR community.  Of course ICLR accepts application papers, but nevertheless, in my view, these papers need to focus on learning representation (in its broadest interpretation) to be of interest to the community.  Very often the applications require the use of novel learning representation.
> >
> > I understand the tension between clarity and generality---it always difficult to describe a very general method while still make the description clear.  It makes it especially difficult when you are introducing concepts that our outside the domain of expertise of the intended audience.  I definitely believe that the work you have done is of value, I still wonder if ICLR is the best venue for the work.
> >
> > I'm sorry, but given the lateness of the response and my other commitments I do not have time to reread your amended paper before the end of the review process.

---

> > > ### Author Response · Authors · 2024-11-27
> > >
> > > Thank you for your quick reply! We apologize for the late response, which is due to the large number of additional experiments that we conducted in the revision. These additional experiments, as well as the significant revision of the introduction (as recommended), allowed us to strengthen the ML content of the paper to be of greater relevance to the ICLR audience.
> > > Given the extended discussion period, we would highly appreciate if you still find the time to re-review the manuscript.

---

### Official Review · Reviewer_3pjs · 2024-10-31

**Soundness:** 3
**Presentation:** 3
**Contribution:** 2
**Rating:** 5
**Confidence:** 4

**Summary:**

Despite Bayesian optimization (BO) being a common method for identifying the best parameter sets for individual tasks, its application in planning experiments to achieve these universal optima is not as prevalent. The authors addresses the real-world challenge of optimizing chemical reaction conditions to assess if a focus on generality in BO can quicken the discovery of universally optimal conditions and if these optima are applicable to new, unseen scenarios. This assessment is conducted by meticulously framing the problem as an optimization of curried functions and by rigorously comparing generality-focused approaches against real-world experimental data. Their findings indicate that the optimization of general reaction conditions is influenced by the way substrates are sampled, with random sampling proving to be more effective than strategies that rely heavily on data.

**Strengths:**

* The paper is well written, and the method is well described.
* I greatly appreciate the author for providing a multitude of real chemical experiment scenarios and testing the algorithm's performance on them, which has given us considerable insight into the practical application of BO.

**Weaknesses:**

* Although I appreciate the extensive work done by the authors, there is a significant issue with this article in Section 2, the problem formulation. In fact, problem (2) is not a new issue in the field of BO, depending on the form of $\phi(\cdot)$. For instance, when $\phi(\cdot)$ takes the form of Mean aggregation, Threshold aggregation, and MSE aggregation, such problems are referred to as Bayesian Optimization with Expensive Integrands[1,2]. When $\phi(\cdot)$ takes the form of Minimum Aggregation, these problems are known as Robust Bayesian optimization [3] (or minimax Bayesian Optimization). Therefore, the statements in section 2.2.2 are not representative, and a substantial revision may be needed for the Section 2.

* Although the authors have compared the BO method with those in the chemistry-related field, given that there are already many good methods in the BO field, these methods should also be evaluated on the chemical dataset. In fact, the conclusions of the article are thus questionable—whether the random selection performs better because it truly has good performance, or just because the comparative methods were not chosen correctly?

[1] Toscano-Palmerin S, Frazier P I. Bayesian optimization with expensive integrands[J]. SIAM Journal on Optimization, 2022, 32(2): 417-444.

[2] Xie J, Frazier P I, Sankaran S, et al. Optimization of computationally expensive simulations with Gaussian processes and parameter uncertainty: Application to cardiovascular surgery[C]//2012 50th annual allerton conference on communication, control, and computing (Allerton). IEEE, 2012: 406-413.

[3] Bogunovic I, Scarlett J, Jegelka S, et al. Adversarially robust optimization with Gaussian processes[J]. Advances in neural information processing systems, 2018, 31.

**Questions:**

I notice that many methods involving random selection tend to show a sudden surge in performance on the figures. What could be the cause of this situation?

---

> ### Author Response · Authors · 2024-11-25
>
> > Although I appreciate the extensive work done by the authors, there is a significant issue with this article in Section 2, the problem formulation. In fact, problem (2) is not a new issue in the field of BO, depending on the form of $\Phi(\cdot)$. For instance, when $\Phi(\cdot)$ takes the form of Mean aggregation, Threshold aggregation, and MSE aggregation, such problems are referred to as Bayesian Optimization with Expensive Integrands [1,2]. When $\Phi(\cdot)$ takes the form of Minimum Aggregation, these problems are known as Robust Bayesian optimization [3] (or minimax Bayesian Optimization). Therefore, the statements in section 2.2.2 are not representative, and a substantial revision may be needed for the Section 2.
> > [1] Toscano-Palmerin S, Frazier P I. Bayesian optimization with expensive integrands[J]. SIAM Journal on Optimization, 2022, 32(2): 417-444.
> > [2]  Xie J, Frazier P I, Sankaran S, et al. Optimization of computationally expensive simulations with Gaussian processes and parameter uncertainty: Application to cardiovascular surgery[C]//2012 50th annual allerton conference on communication, control, and computing (Allerton). IEEE, 2012: 406-413.
> > [3] Bogunovic I, Scarlett J, Jegelka S, et al. Adversarially robust optimization with Gaussian processes[J]. Advances in neural information processing systems, 2018, 31.
>
> Thank you for pointing out the connection to BO of expensive integrands and DRBO. In line with the recommendations, we have largely restructured the “related works” section and emphasized the connections to stochastic optimization (BO with expensive integrands) and DRBO. Based on the reviewer’s recommendations, we have also extended the discussion of the motivation of formulating the problem as BO over curried functions (Section 2.2.4).
>
> > Although the authors have compared the BO method with those in the chemistry-related field, given that there are already many good methods in the BO field, these methods should also be evaluated on the chemical dataset. In fact, the conclusions of the article are thus questionable—whether the random selection performs better because it truly has good performance, or just because the comparative methods were not chosen correctly?
>
> Following the reviewer’s recommendations, we performed a series of additional experiments. We provide two additional benchmark problems, and systematically evaluate a series of BO methods on all four tasks. For example, we implemented and evaluated BO approaches similar to those reported by Toscano-Palmerin and Frazier [1] (i.e., joint optimization in $\mathcal{X} \times \mathcal{W} space using a two-step lookahead acquisition function, referred to as JOINT 2LA-EI in the manuscript). This allows us to perform a more comprehensive analysis to identify effective strategies for generality-oriented optimization.
>
> > I notice that many methods involving random selection tend to show a sudden surge in performance on the figures. What could be the cause of this situation?
>
> Thank you for pointing this out! We noticed that these performance jumps could be traced back to slight numerical instabilities in calculating the GAP metric and visualizing the optimization trajectories, which have been fixed.

---

### Official Review · Reviewer_Hzw2 · 2024-11-03

**Soundness:** 2
**Presentation:** 3
**Contribution:** 2
**Rating:** 3
**Confidence:** 4

**Summary:**

This paper studies Bayesian optimization (BO) for general parameters discovery in chemistry, especially formulating the problem as a generality-oriented optimization, where the authors aim to identify general reaction conditions that perform well across diverse substrates. They benchmark the recent algorithms and discuss the difficulty of applying BO for this scenario.

**Strengths:**

1. This work focuses on general parameters discovery in chemistry, which is a significant scientific problem.
2. This work proposes a well-established benchmark for this problem.
3. The code implementation is well-structured and clear.

**Weaknesses:**

1.	Lack of discussion with related topics. The formulation of generality-oriented optimization is quite similar to the definition of stochastic optimization [1] and distributionally robust optimization, with the context variables (i.e., substrates in this paper) sampled from a distribution determined by the aggregation function $\Phi$, and the framework of generality-oriented BO aligns with the simulator setting of stochastic/distributionally robust BO [2, 3, 4, DRBO]. However, this work lacks a systematic discussion and an experimental comparison between generality-oriented BO and stochastic/distributionally robust BO.

2.	Weak experimental analysis and discussion. While the authors aim to investigate the challenges in optimizing general parameters, as indicated by the title, their conclusion attributes these difficulties primarily to the substrate sampling strategy. However, this conclusion appears to be drawn from a limited comparison between random acquisition and posterior variance acquisition of \textbf{w}_{\textrm{next}} under a single fixed strategy of \textbf{x}_{\textrm{next}}. A more comprehensive analysis would involve comparing various acquisition policies for \textbf{w}_{\textrm{next}} across different effective strategies of \textbf{x}_{\textrm{next}}. Such strategies could include UCB with varying $\beta$, or pure exploration for continuous/discrete/mixed-variable spaces, as demonstrated in Vizier Bandit [5]. To conclude, I will be glad if the authors could add a discussion on the impact between acquisition policies for both \textbf{w}_{\textrm{next}} and \textbf{x}_{\textrm{next}}.


3.	Limited insights into the design of BO algorithms. As an AI paper, it would be better to offer broader insights beyond introducing the problem. Unfortunately, the work is mainly an application of BO methods, making the algorithmic novelty somewhat limited.

4. The paper contains some minor language problems, including

- line 44, “efficiency. (Clayton et al. 2019; …).” → “efficiency (Clayton et al. 2019; …).”

- line 173, “thresholdBetinol et al. (2023)” → “threshold (Betinol et al. 2023)”

- line 265, “work in this field” → “works in this field”

- inconsistent tenses in Section 3 and Section 4, where Section 3.1 uses present tense while the remaining sections use past tense

- line 490, “RDKit: Open-source cheminformatics” → “Greg Landrum. RDKit: Open-source cheminformatics”

- inconsistent capitalization style of section headings. The paper uses title case only in Appendix A.1 and its subsections (A.1.1, A.1.2), while all other sections follow sentence case format

- line 850, “at time point $k$ $p(g_k(\mathbf{x}))$” → “at time point $k$, $p(g_k(\mathbf{x}))$”

- line 892, “SAA” lacks a citation

- line 1278, a citation of GPyTorch is needed

- line 1293, “Bandit Wang et al. (2024)” → “Bandit (Wang et al. 2024)”

- line 1403, “substrates for chosen for” → “substrates chosen for”

- line 1565, “, that” → “, which”, “(i.e. above)” → “(i.e., above)”

[1] Toscano-Palmerin and Frazier. Bayesian Optimization with Expensive Integrands. SIAM Journal on Optimization, 2022.
[2] Kirschner et al. Distributionally Robust Bayesian Optimization. AISTATS, 2020.
[3] Nguyen et al. Distributionally Robust Bayesian Quadrature Optimization. AISTATS, 2020.
[4] Husain et al. Distributionally Robust Bayesian Optimization with $\varphi$-divergences. NeurIPS, 2023.

**Questions:**

1.	As in weakness addressed, can the author provide discussion or experimental comparison with stochastic/distributionally robust BO?

2.	In Figure 3, the efficiency of augmented search space is not so trivial, especially for the N.S-Acetal formation, and it will be better if the authors could show the improvement of Spearman correlation coefficient.

3.	Since the authors have shown that the better transferability of the augmented search space, why not just provide the results on the augmented search space? What is the meaning of providing results on the original search space?

4.	In line 419, what is the full name and the nature of the metric GAP? E.g., its scale and is it better to maximize it?

5.  Is the pretrained random forest regressor a reasonable oracle? Except for MAE and MSE metric in Appendix A.2.2, metrics like Spearman correlation coefficient on the held-out test set should also be shown, as the random forest regressor for Superconductor task in Design-Bench [1].

I am more than happy to increase my score if the authors can address my questions.

[1] Trabucco et al. Design-Bench: Benchmarks for Data-Driven Offline Model-Based Optimization. ICML 2022.

---

> ### Author Response · Authors · 2024-11-25
>
> > Lack of discussion with related topics. The formulation of generality-oriented optimization is quite similar to the definition of stochastic optimization [1] and distributionally robust optimization, with the context variables (i.e., substrates in this paper) sampled from a distribution determined by the aggregation function $\Phi$, and the framework of generality-oriented BO aligns with the simulator setting of stochastic/distributionally robust BO [2, 3, 4, DRBO]. However, this work lacks a systematic discussion and an experimental comparison between generality-oriented BO and stochastic/distributionally robust BO.
> >
> > [1] Toscano-Palmerin and Frazier. Bayesian Optimization with Expensive Integrands. SIAM Journal on Optimization, 2022.
> >[2] Kirschner et al. Distributionally Robust Bayesian Optimization. AISTATS, 2020.
> > [3] Nguyen et al. Distributionally Robust Bayesian Quadrature Optimization. AISTATS, 2020.
> > [4] Husain et al. Distributionally Robust Bayesian Optimization with -divergences. NeurIPS, 2023.
>
> We sincerely thank the reviewer for their extremely helpful and constructive comments! In line with their recommendations, we have largely restructured Section 2.2.4 and emphasized the connections to stochastic optimization (BO with expensive integrands) and DRBO. Based on the reviewer’s recommendations, we have also extended the discussion of the motivation of formulating the problem as BO over curried functions.
>
> Moreover, we performed a series of further benchmark experiments. We provide two further benchmark problems, as well as a systematic analysis of different BO strategies on all four benchmark tasks: As suggested by the reviewer, we implemented and evaluated BO approaches similar to those reported by Toscano-Palmerin and Frazier [1] (i.e., joint optimization in $\mathcal{X} \times \mathcal{W} space using a two-step lookahead acquisition function, referred to as JOINT 2LA-EI in the manuscript).
>
> These experiments allow us to place and discuss our findings in the context of the works raised by the reviewer.
>
> > Weak experimental analysis and discussion. While the authors aim to investigate the challenges in optimizing general parameters, as indicated by the title, their conclusion attributes these difficulties primarily to the substrate sampling strategy. However, this conclusion appears to be drawn from a limited comparison between random acquisition and posterior variance acquisition of wnext under a single fixed strategy of xnext. A more comprehensive analysis would involve comparing various acquisition policies for wnext across different effective strategies of xnext. Such strategies could include UCB with varying $\beta$, or pure exploration for continuous/discrete/mixed-variable spaces, as demonstrated in Vizier Bandit [5]. To conclude, I will be glad if the authors could add a discussion on the impact between acquisition policies for both wnext and xnext.
>
> Thank you for this suggestion! In response, we have fully restructured the “Results and Discussion” section of our chapter, now providing further experimental results, as well as a systematic discussion. We first discuss different strategies for acquiring $x_{next}$, followed by a systematic analysis of different acquisition functions for selecting $w_{next}$. Eventually, we compare sequential x–w and joint x/w acquisition policies for finding general optima. We believe that these additional experiments, along with the inclusion of two further benchmark tasks, provide more insightful conclusions on the choice of strategies for generality-oriented optimization.
>
> > Limited insights into the design of BO algorithms. As an AI paper, it would be better to offer broader insights beyond introducing the problem. Unfortunately, the work is mainly an application of BO methods, making the algorithmic novelty somewhat limited.
>
> We thank the reviewer for emphasizing this weakness of our initial submission. We would like to emphasize that the objective of the paper is to provide a benchmark of different strategies towards generality-oriented optimization. While we formulate the problem as BO over curried function – merely to gain the flexibility that is required to incorporate practically relevant generality metrics – the goal is not to provide algorithmic novelty. Therefore, we have substantially revised our manuscript to make this message come across clearer.  We are confident that the implemented changes provide a more comprehensive benchmark to achieve that goal and thus make this paper suitable for publication at ICLR.
>
> > The paper contains some minor language problems, including
>
> All problems have been addressed in the revised version.

---

> ### Author Response · Authors · 2024-11-25
> **Answers to questions**
>
> > As in weakness addressed, can the author provide discussion or experimental comparison with stochastic/distributionally robust BO?
>
> We want to thank you again for pointing out the relevance of stochastic BO and DRBO. In accordance with your comments, we provide a discussion on the relation of BO over curried functions to stochastic and distributionally robust BO. Further, as we benchmark results for mean and threshold aggregation in this paper, we provide additional experimental  comparison of the algorithm proposed by Toscano-Palmerin and Frazier [1] (referred to as JOINT 2LA-EI in the manuscript).
>
> > In Figure 3, the efficiency of augmented search space is not so trivial, especially for the N.S-Acetal formation, and it will be better if the authors could show the improvement of Spearman correlation coefficient.
>
> We added the Spearman rank correlation coefficient in the plot of Figure 3 for all four benchmarks and expanded on the discussion of the utility of generality-oriented BO (Section 4). As elaborated on in the main text, the focus lies on the achievement of higher overall generality scores through the data augmentation, underlying the necessary modifications to established benchmarks to reflect real-world scenarios.
>
> > Since the authors have shown that the better transferability of the augmented search space, why not just provide the results on the augmented search space? What is the meaning of providing results on the original search space?
>
> Following your suggestion, we have largely restructured the main text, incorporated further experimental findings, and discuss the benchmarked algorithm performance on the augmented benchmarks with two aggregation metrics, namely the mean aggregation and the threshold aggregation. The results on the original benchmarks are provided in the appendix (A.6.3). This leaves more room for a detailed discussion of the observed trends and nuanced effects.
>
> > In line 419, what is the full name and the nature of the metric GAP? E.g., its scale and is it better to maximize it?
>
> We have added the equation describing the calculation of GAP and a description of its conceptualization as a normalized (i.e. between 0 and 1), problem-independent metric (see Section 3.2). However, to the best of our knowledge, GAP is not an abbreviation, but the metric has been reported and utilized under this name (see citation Jiang et al., 2020)
>
> > Is the pretrained random forest regressor a reasonable oracle? Except for MAE and MSE metric in Appendix A.2.2, metrics like Spearman correlation coefficient on the held-out test set should also be shown, as the random forest regressor for Superconductor task in Design-Bench [1].
> [1] Trabucco et al. Design-Bench: Benchmarks for Data-Driven Offline Model-Based Optimization. ICML 2022.
>
> Thank you for pointing this out! We have also added the evaluation of the random forest regressor in a five-fold cross validation and discussed the results in Appendix A.2.2.

---

### Official Review · Reviewer_fCVf · 2024-11-11

**Soundness:** 3
**Presentation:** 2
**Contribution:** 2
**Rating:** 3
**Confidence:** 3

**Summary:**

This paper investigates the setting of Bayesian optimization where some variables can be manipulated as decision variables, and others must be optimized over in some aggregate measure, providing a measure of generality. This is motivated by the challenge of optimizing chemical reactions over general parameters (i.e., finding reaction conditions that perform well for multiple substrates), an area that is beneficial but underexplored in practical applications. The authors compare several Bayesian Optimization (BO) methods to identify reaction conditions that can be effectively applied across a range of simulated chemical reactions, giving a sense of the practical challenges involved in this task. The work emphasizes the difficulty of generality-oriented BO due to the partial observability of results (evaluations can only be performed on singletons or subsets of possible substrates), necessitating innovative BO algorithms.

**Strengths:**

- The problem formulation and motivation are presented well. The approach of using curried functions to define the setting of generality-oriented optimization adds some mathematical clarity and supports further research in this direction, though I am unsure how much of this is novel to this work.
- The use of real-world datasets and extensive comparisons across various algorithms, including recently proposed algorithms, provide good insights into the current challenges in the area of generality-oriented BO and the effectiveness of currently available strategies.

**Weaknesses:**

- My main concern with this paper is its suitability for ICLR. In terms of length, many of the details required to understand the paper are moved to the appendix, making the paper difficult to read. In terms of content, many of the details concern practical challenges related specifically to the application of chemical reaction engineering (is this method applicable in ML or other domains?). For example, how to modify the simulated chemistry benchmarks to better suit this exact domain problem setting. For both reasons, this paper is likely better suited as a full-length journal article in chemistry, chemical engineering, or data-driven engineering.
- A related concern is that, while the domain-specific elements are interesting and discussed in detail, the ML elements are not, leaving many open questions (especially without the appendices, which reinforces the above point). For example, the authors find that randomizing the selection of elements within $w$ to sample at works well. This suggests that the surrogate model of $w$ used to select optimal elements is ineffective, or the prior distribution is wrong. The selection of how to build a statistical model over the unordered elements of should be discussed in detail so that the reader can understand the approach taken.

**Questions:**

-	Regarding the connection to multiobjective optimization, is the case considered here equivalent to scalarizing a multiobjective problem, where all objectives are given equal weights (thus producing the mean)?
-	Similarly, does not the choice of $m<n$ elements of $w$ to sample effectively produce a multi-fidelity setting?

---

> ### Author Response · Authors · 2024-11-25
>
> > My main concern with this paper is its suitability for ICLR. In terms of length, many of the details required to understand the paper are moved to the appendix, making the paper difficult to read. In terms of content, many of the details concern practical challenges related specifically to the application of chemical reaction engineering (is this method applicable in ML or other domains?). For example, how to modify the simulated chemistry benchmarks to better suit this exact domain problem setting. For both reasons, this paper is likely better suited as a full-length journal article in chemistry, chemical engineering, or data-driven engineering.
> >
> > A related concern is that, while the domain-specific elements are interesting and discussed in detail, the ML elements are not, leaving many open questions (especially without the appendices, which reinforces the above point). For example, the authors find that randomizing the selection of elements within $w$ to sample at works well. This suggests that the surrogate model of $w$ used to select optimal elements is ineffective, or the prior distribution is wrong. The selection of how to build a statistical model over the unordered elements of should be discussed in detail so that the reader can understand the approach taken.
>
> We thank the reviewer for their comments. In line with their suggestions, we have expanded on the algorithmic considerations when performing BO over curried functions in the main text (section 2.2.4). This provides greater understanding of the methods within this paper and further shifts the focus on the ML content of the paper. In addition, we added two further benchmark datasets (Appendix A.2) and conducted extensive additional experiments (section 4)  to provide a more systematic investigation and discussion on the algorithmic requirements for successful generality-oriented optimization.
>
> We are confident that with these modifications, the publication is of higher interest to the ML community at ICLR. In addition we want to point out that the Call for Papers explicitly states “applications to physical sciences (physics, chemistry, biology, etc.) as a relevant topic for publication at this conference.
>
> We thank the reviewer again for their helpful comments and hope that the reviewer agrees that with these modifications, the publication is suitable for acceptance at ICLR.
>
>
> > Regarding the connection to multiobjective optimization, is the case considered here equivalent to scalarizing a multiobjective problem, where all objectives are given equal weights (thus producing the mean)?
>
> We agree with the reviewer that, in the case of the mean aggregation, the function to optimize is mathematically equivalent to a scalarized multi-objective problem, where all objectives are equally weighted. In our case, however, the partial monitoring of the objective remains, which is not the case for conventional multi-objective optimization. As such, the problem is rather related to Bayesian Optimization with expensive integrands, as emphasized by Reviewers Hzw2 and 3pjs, and we have updated the main text of our paper to highlight these similarities (section 2.2.4).
>
> We have decided on the formulation as optimization over a curried function, as this allows us to provide a flexible framework for accommodating different kinds of aggregation methods. We illustrate this in the revised manuscript by providing systematic analyses for both the mean and the threshold aggregation functions.
>
>
> > Similarly, does not the choice of $m < n$ elements of $w$ to sample effectively produce a multi-fidelity setting?
>
> The setting of $m < n$ leads to a partial monitoring scenario, i.e. the true objective (potentially with noise) is observed, albeit not fully. In contrast, in the typical multi-fidelity setting, only approximations to the true objective are observed, at varying costs. We have emphasized the distinction of generality-oriented optimization from multiobjective and multifidelity optimization in section 2.2.3.

---

### Author Response · Authors · 2024-12-02

Good day!

As the deadline for reviewer comments is rapidly approaching (end of today, AoE), we kindly wanted to ask you if you had the opportunity to consider our changes to the manuscript, including the extensive additional experiments; We thank you for your time and highly appreciate your feedback!

---

### Meta-Review · Area_Chair_LU4d · 2024-12-20

**Metareview:**

Thank you for your submission to ICLR. This paper focuses on Bayesian optimization over curried functions, to find so-called “general optima” that are optimal parameters for multiple tasks. The authors develop an acquisition function and Bayesian optimization algorithm for this setting, focusing on the problem of chemical reaction condition optimization. While reviewers appreciate the mathematical formalism and experimental benchmark, they still have a number of remaining concerns, involving the paper organization and clarity of presentation, connections with and relations to prior work, and lack of experimental results or comparisons against sufficient baseline methods. I encourage the authors to carefully consider and incorporate these comments from reviewers upon resubmission.

**Additional Comments On Reviewer Discussion:**

During the rebuttal period, the authors responded back to most questions posed by reviewers and made a number of updates in their paper. However, the reviewers remained unconvinced, and after internal discussion, they maintained their scores.

---

### Decision · Program_Chairs · 2025-01-22

Reject